# IgG and IgM cooperate in coating of intestinal bacteria in IgA deficiency

Carsten Eriksen[1,2], Janne Marie Moll [1], Pernille Neve Myers [1],
Ana Rosa Almeida Pinto [1], Niels Banhos Danneskiold-Samsøe [3],
Rasmus Ibsen Dehli [1], Lisbeth Buus Rosholm[1], Marlene Danner Dalgaard[4],
John Penders [5], Daisy MAE Jonkers [6], Qiang Pan-Hammarström [7],
Lennart Hammarström [7], Karsten Kristiansen [2,3,8,9] & Susanne Brix [1,2] ✉

Immunoglobulin A (IgA) is acknowledged to play a role in the defence of the mucosal barrier by coating microorganisms. Surprisingly, IgA-deficient humans exhibit few infection-related complications, raising the question if the more specific IgG may help IgM in compensating for the lack of IgA. Here we employ a cohort of IgA-deficient humans, each paired with IgA-sufficient household members, to investigate multi-Ig bacterial coating. In IgA-deficient humans, IgM alone, and together with IgG, recapitulate coating of most bacterial families, despite an overall 3.6-fold lower Ig-coating. Bacterial IgG coating is dominated by IgG1 and IgG4. Single-IgG2 bacterial coating is sparse and linked to enhanced *Escherichia coli* load and TNF-α. Although single-IgG2 coating is 1.6-fold more prevalent in IgA deficiency than in healthy controls, it is 2-fold less prevalent than in inflammatory bowel disease. Altogether we demonstrate that IgG assists IgM in coating of most bacterial families in the absence of IgA and identify single-IgG2 bacterial coating as an inflammatory marker.

The intestine constitutes the largest interacting interface between microbes colonizing our body and the host immune system. In order to preserve a homeostatic non-inflammatory environment in the human body, it is important to maintain a distinct compartmentalization between microbes and adjacent tissues. Immunoglobulin A (IgA) takes part in this first line of defense and binds both commensal and pathogenic bacteria[1]. IgA is believed to be produced mainly as a direct response to intestinal colonization with microbes[2], and is highly abundant in the intestinal mucosal areas where more than 80% of antibody-secreting plasma cells are located[3].

IgA deficiency in humans is relatively common with a prevalence of 1/600 in Caucasians[4]. Most IgA deficient subjects remain asymptomatic with only a mild phenotype[4]. Early studies suggested that IgM to some extent may compensate for the lack of IgA[5], and it is now well established that IgA[6–9] and the less specific IgM[6,8,10] bind members of the gut microbiota in healthy individuals. Recently it was also shown that IgG, which generally binds with a higher affinity than IgM, coats

[1]Department of Biotechnology and Biomedicine, Technical University of Denmark, Kgs. Lyngby, Denmark. [2]Center for Molecular Prediction of Inflammatory Bowel Disease, Department of Clinical Medicine, Aalborg University, Copenhagen, Denmark. [3]Laboratory of Genomics and Molecular Biomedicine, Department of Biology, University of Copenhagen, Copenhagen, Denmark. [4]Department of Health Technology, Technical University of Denmark, Kgs. Lyngby, Denmark. [5]Department of Medical Microbiology, Infectious Diseases and Infection Prevention, NUTRIM School for Nutrition and Translational Research in Metabolism & Care and Public Health Research Institute CAPHRI, Maastricht University Medical Centre, Maastricht, The Netherlands. [6]Division Gastroenterology-Hepatology, Department of Internal Medicine, NUTRIM School for Nutrition and Translation Research in Metabolism, Maastricht University Medical Centre+, Maastricht, The Netherlands. [7]Division of Immunology, Department of Medical Biochemistry and Biophysics, Karolinska Institutet, Stockholm, Sweden. [8]BGI-Shenzhen, Shenzhen, China. [9]Qingdao-Europe Advanced Institute for Life Sciences, Qingdao, Shandong, China. ✉e-mail: sbrix@dtu.dk

gut bacteria[11]. Moreover, it was reported that the gut microbiota is able to promote serum IgG levels[12], and that these IgGs can bind to gut bacteria when binding is examined in vitro[13]. However, it is yet unclear to what extent IgM and IgG interplay in in vivo coating of bacteria in the absence of IgA, and how coating with the subtypes of IgG, IgG1, IgG2, IgG3, or IgG4 is manifested.

Previous studies addressing the influence of IgA-coating of gut bacteria have compared data from IgA deficient subjects with a general non-related IgA+ control group[6] that does not allow for control of lifestyle differences, which may impact the gut microbiota composition and thereby possibly influence Ig-coating patterns. Household members have more similar gut microbiota composition than people from different households[14,15], a setup comparing data (IgA−) and IgA sufficient (IgA+) household members is necessary to reduce the impact of differences in lifestyle between IgA− and IgA+ individuals[15].

Here we use quantitative and relative profiling of gut bacterial multi-Ig coating with IgA, IgM. IgG, including IgG1–4 and 16 S rRNA gene amplicon sequencing of sorted Ig-coated bacteria are employed to identify the number and nature of coated bacterial taxa. Publicly available data are next used to generate a map of the anatomical location of coated bacteria within the human gut, and we perform in silico pathway analysis to predict if highly coated bacteria possess specific immunostimulatory ligands known to promote specific Ig production. Our results thus accurately describe the barrier defense differences that characterize IgA-deficient humans, and explain why they experience relatively mild disease symptoms.

## Results

### Ig-coating of gut bacteria in IgA+ and IgA- subjects

We first performed a simultaneous phenotyping of IgA, IgM, and IgG coating of gut bacteria from IgA deficient subjects (IgA−, $n = 31$) and their IgA sufficient household members (IgA+, $n = 31$) using multi-color flow cytometry-based analysis of fecal bacteria (Fig. 1a). IgA deficiency was confirmed by the lack of IgA coating of gut bacteria (Fig. 1b). In IgA+, we identified a median relative IgA coating of gut bacteria of 8.93% (Fig. 1b, Supplementary Table 1). IgM coating of gut bacteria was higher in IgA− (2.18%) compared to IgA+ subjects (0.17%, $P = 1.5 \times 10^{-4}$, Fig. 1b, Supplementary Table 1), while IgG coating was similar in IgA− (1.17%) and IgA+ subjects (2.21%, $P = 0.11$, Fig. 1b, Supplementary Table 1). This finding of a sizable fraction of IgG coated gut bacteria in healthy adults contrasts a previously published report of minute levels of bacterial IgG coating in humans (median: 0.03%)[13], but is in line with results obtained in studies of healthy human adults[16] and infants[17], and reported in mice[12].

We quantified the actual number of bacteria/g feces using flow cytometry counting, and found no differences in the total gut bacteria number between IgA+ and IgA− subjects ($P = 0.8$, Fig. 1c, Supplementary Table 1). Based on this, and the percentage distribution provided above, it was evident that the number of IgG coated gut bacteria was on average only 3.7-fold lower than for IgA in IgA+ subjects, and at a scale similar to that of IgM coated bacteria in IgA− subjects (Fig. 1d, Supplementary Table 1).

An inverse correlation between the number of gut bacteria/g feces and the % of IgA coated bacteria in IgA+ subjects was observed (SCC = −0.39, $P = 0.03$, Fig. 1e), indicative of a stable production of IgA during

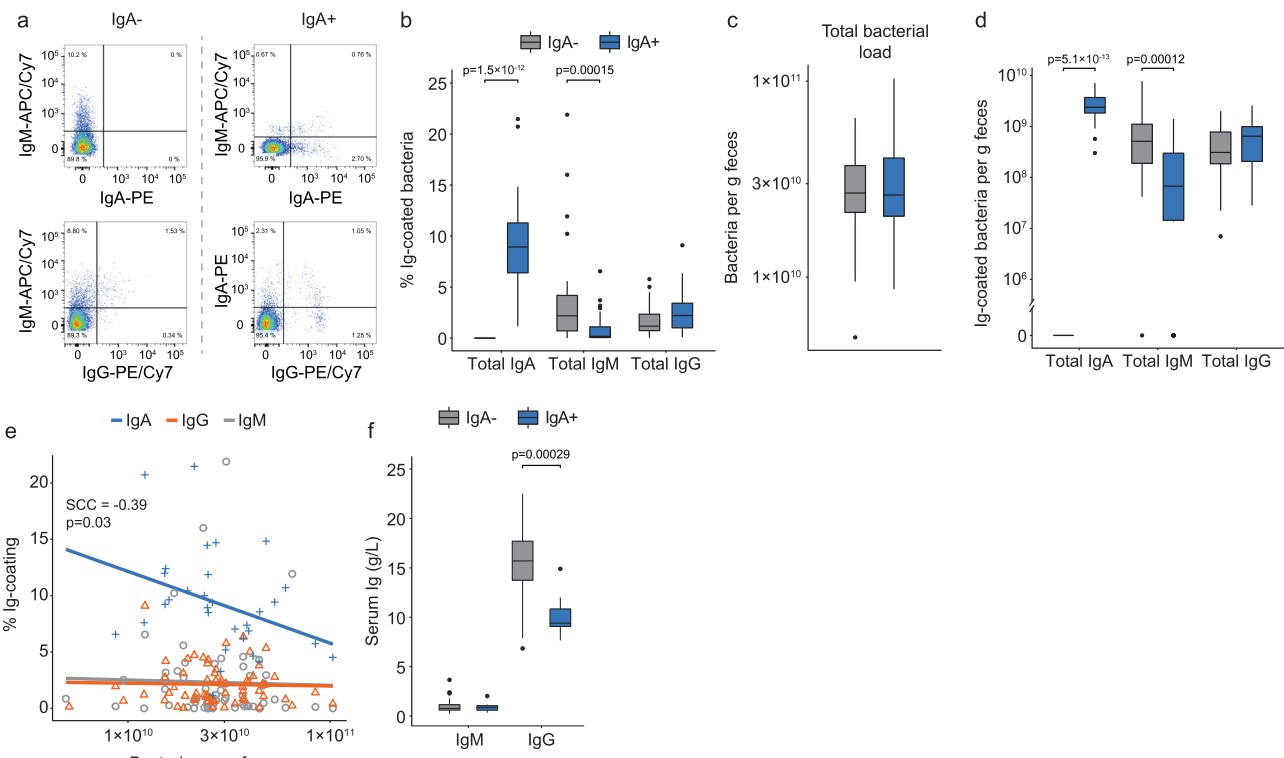

**Fig. 1 | Fecal bacterial Ig-coating in the presence and absence of IgA.**
**a** Representative flow cytometry-based analysis of fecal bacteria coated with IgA, IgM and IgG in IgA deficient subjects (IgA−, $n = 31$, left panel) and IgA+ household members (IgA+, $n = 31$, right panel). **b** Relative % of IgA, IgM and IgG coated fecal bacteria in IgA− ($n = 31$) and IgA+ ($n = 31$) subjects. **c** Total bacterial load ($n = 2 \times 31$). **d** number of Ig-coated fecal bacteria/g feces ($n = 2 \times 31$). **e** Load of bacteria in feces vs. relative % of Ig-coated fecal bacteria tested using Spearman's correlation (IgA: $n = 31$, IgG and IgM: $n = 2 \times 31$). The straight lines indicate the parametric

correlation coefficient. **f** Serum levels (g/L) of IgG and IgM in IgA− ($n = 31$) and IgA+ ($n = 10$) subjects. A grey fill is used to indicate IgA− subjects, while a blue fill indicates IgA+ subjects. Paired Wilcoxon test was used to compare IgA− and IgA+ pairs. $P$ values are two-sided and adjusted using FDR. For boxplots, the center line indicates the median and the box limits indicate the quartiles. Whiskers extend to the data points within 1.58× the interquartile range, and outliers are shown as individual dots.

homeostatic conditions which is not increasing with increasing gut bacterial load. By contrast, the % of bacteria with IgM or IgG coating remained constant regardless of gut bacterial load (Fig. 1e), suggesting that production of IgM and IgG is dependent on the gut bacterial load in humans, as reported for mice[18]. Despite similar gut bacterial loads in IgA- vs. IgA+ subjects, and a switch to gut bacterial coating with both IgM and IgG in the absence of IgA, we identified elevated circulating IgG in IgA− vs. IgA+ subjects ($P = 2.9 \times 10^{-4}$, Fig. 1f, Supplementary Table 1), which is suggestive of a less well guarded barrier in IgA deficient humans. We therefore focused to define the different dynamics in gut bacterial coating in the presence and absence of IgA.

## Dual IgM and IgG gut bacterial coating in absence of IgA

Using the multi-color Ig-coating data, we next examined whether gut bacteria were simultaneously coated with one (IgA, IgG or IgM) or multiple antibodies. We consistently detected non-coated bacteria, bacteria coated with a single antibody isotype (single-coating), with two different isotypes (double-coating), and even with all three isotypes in IgA+ subjects (triple-coating). Single-coating of bacteria with IgA was the dominant feature in IgA+ subjects, coating on average $1.5 \times 10^9$ bacteria/g feces, and double-coating with IgA and IgG was the second most prevalent coating type, coating on average $3.2 \times 10^8$ bacteria/g feces. In contrast, single IgM coated on average $2.2 \times 10^8$ bacteria/g feces in IgA− subjects, and double-coating with IgG and IgM was seen in on average $1.8 \times 10^8$ bacteria/g feces in IgA− subjects (Fig. 2a, b, Supplementary Table 2). Single-IgG coated bacteria were

prevalent in both, although slightly higher in IgA− subjects ($P = 0.045$). Triple-coating of bacteria was also evident in IgA+ subjects, coating on average $5.3 \times 10^7$ bacteria/g feces, and identified amongst 61% of IgA+ subjects (Fig. 2a, b). Altogether, we identify single-, double-, or triple-coating of on average $1.9 \times 10^9$ bacteria/g feces (median: 7.2% of all bacteria) in IgA+ subjects and single-, or double-coating of on average $0.55 \times 10^9$ bacteria/g feces (median: 2.0% of all bacteria) in IgA− subjects. This suggests a less efficient microbiota coating, even by a combination of IgM and IgG, when IgA is lacking.

## Partly overlapping bacterial Ig-coating in IgA− vs. IgA+ pairs

Since any IgA and any IgM coating were the major discriminating coating types amongst IgA+ and IgA− subjects, we next used a FACS-based sorting approach[7] to sort out all IgA coated bacteria in IgA+ subjects and all IgM coated bacteria in IgA− subjects to identify the taxonomy and coating frequency of single- and multi-coated bacteria with any IgA or IgM, respectively. We performed the analysis on 16 household pairs of IgA− and IgA+ from which we were able to sort ~$2 \times 10^6$ bacteria per sample. We also sorted out bacteria that were triple-negative for IgA, IgM and IgG (non-coated) from each of these subjects to specifically define coated and non-coated bacteria per subject. Bacteria were identified based on amplicon sequence variants (ASV) analysis of 16 S rRNA gene amplicon sequencing data that were pre-processed by removing reads that appeared either in the pre-sorting fluid, in the blank DNA extraction controls or in the PCR controls (Supplementary Data 1), followed by filtering out ASVs with both

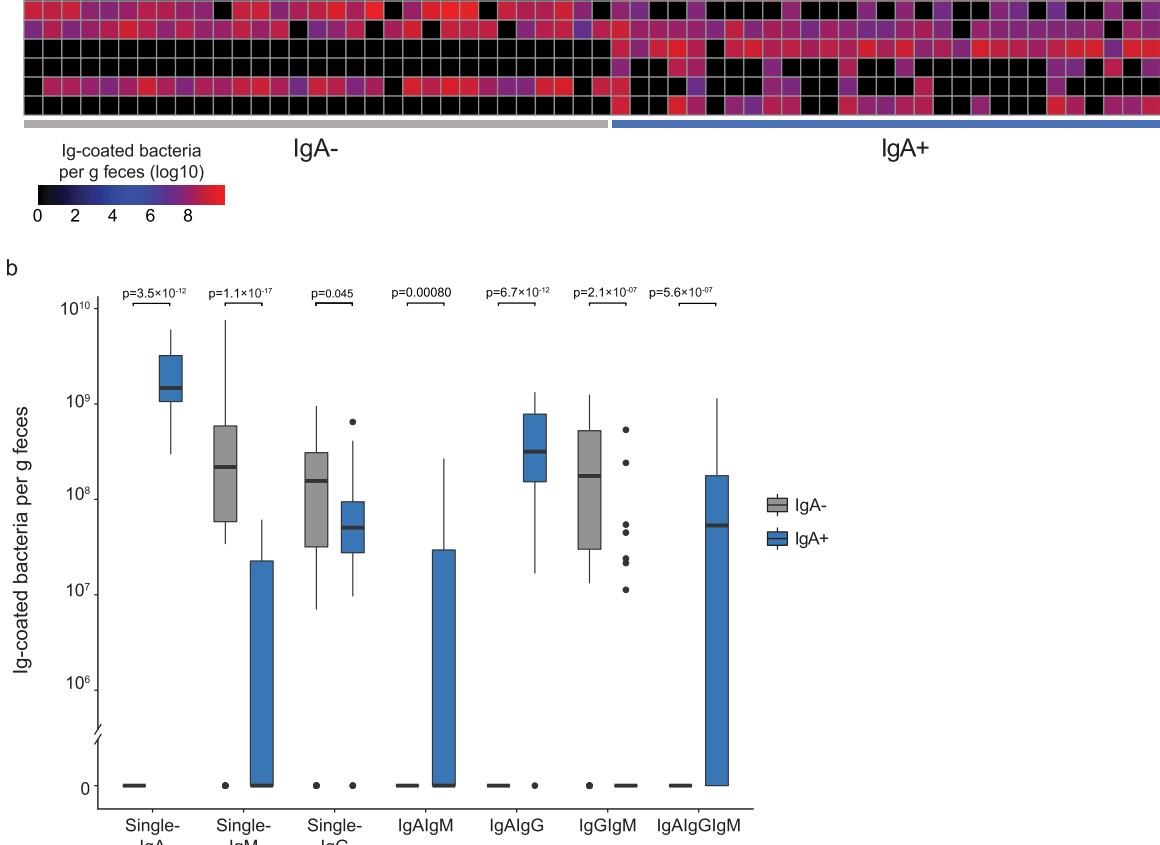

**Fig. 2 | Frequency of single-, double- and triple-Ig coated fecal bacteria in the presence and absence of IgA. a** The individual number of IgA, IgM and/or IgG coated fecal bacteria in IgA− and IgA+ subjects. **b** Average distribution of coated gut bacteria (in gram per feces) in IgA+ and IgA− subjects, $n = 31$ in each group. Paired Wilcoxon test was used to compare IgA− and IgA+ pairs. *P* values are two-sided and adjusted using FDR. A grey fill indicates IgA− subjects, while a blue fill indicates IgA+ subjects. For boxplots, the center line indicates the median and the box limits indicate the quartiles. Whiskers extend to the data points within 1.58x the interquartile range, and outliers are shown as individual dots.

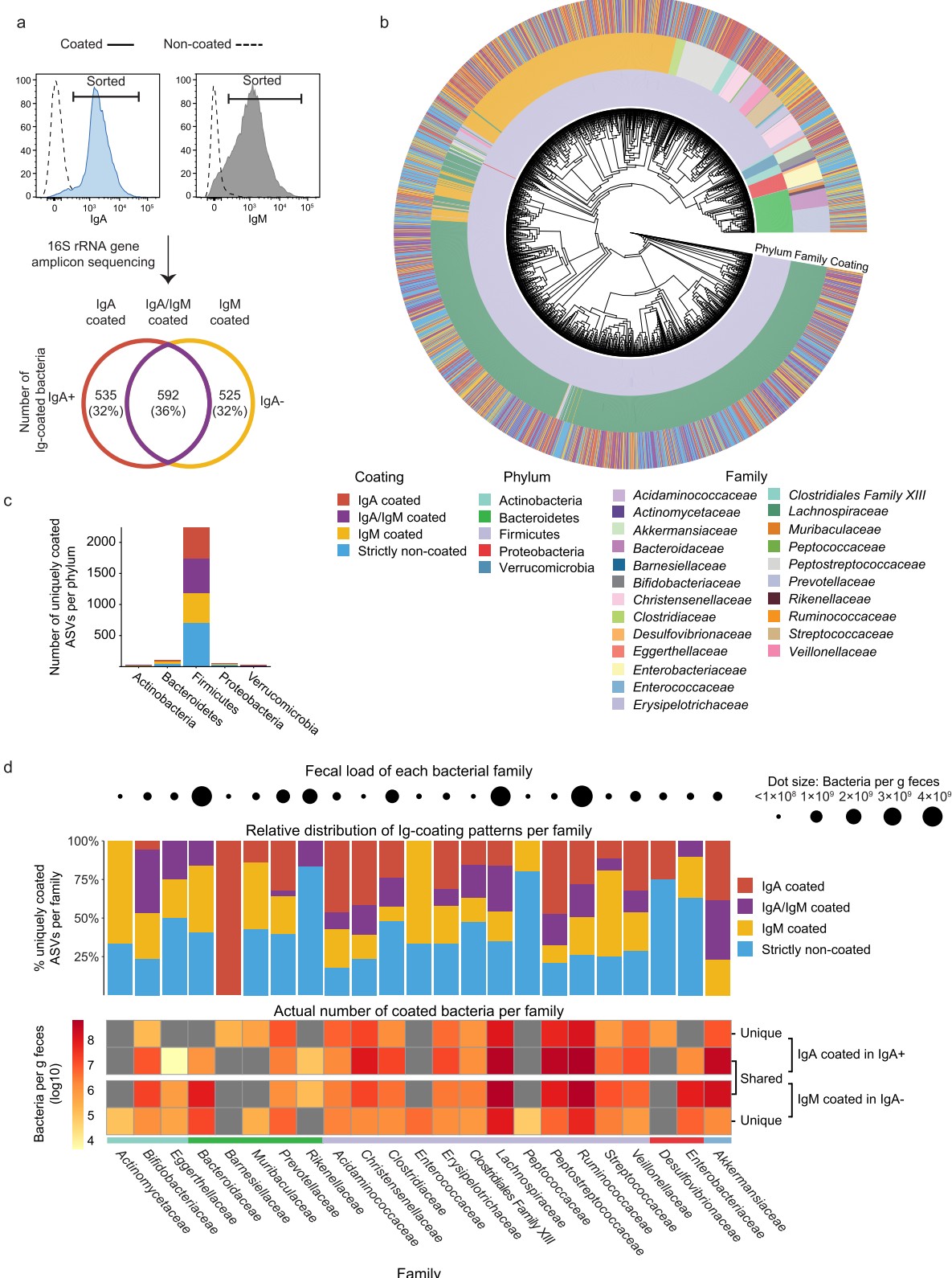

low prevalence and abundance. Based on this, we identified 1652 unique and high-confidence bacteria to be single- or multi-coated with IgA and/or IgM (Fig. 3a, Supplementary Data 2). 1766 bacteria were identified to be non-coated (Supplementary Data 3). We found a partial overlap of 36% of the coated gut bacteria (592 ASVs) that were either single- or multi-coated bacteria with IgA in IgA+ subjects and with IgM

in IgA− subjects (IgA/IgM, Fig. 3a, b, Supplementary Data 2). Of the remaining coated bacteria, 32% (535 ASVs) were identified to be single- or multi-coated with IgA (IgA coated), while 32% of the bacteria (525 ASVs) were single- or multi-coated with IgM (IgM coated) (Fig. 3a, b, Supplementary Data 2). The coating of distinct taxa as well as the number of coated bacteria varied between subjects, and we found

**Fig. 3 | Identification of Ig-coated fecal bacteria. a** All IgA coated and IgM coated bacteria (single- and multi-coated) were isolated via FACS sorting from household-paired samples of IgA+ (*n* = 16) and IgA− subjects. Coated bacteria from both IgA+ and IgA− subjects were sorted and characterized via 16 S rRNA gene amplicon sequencing identifying uniquely IgA coated and IgM coated bacteria based on amplicon sequence variants (ASVs). **b** A phylogenetic tree based on the V3-V4 region of the 16 S rRNA gene from the sequenced IgA coated and IgM coated bacteria. Each of the outer nodes represents a unique bacterium. The inner circle indicates the phylum level of each bacterium, the middle ring the family level and the outer ring indicates which sorting population the bacterium was found in. **c** Number of coated bacteria across bacterial phyla. **d** The heatmap in the lower panel displays the average actual number of Ig-coated bacterial families per g feces in IgA+ and IgA− subjects, respectively. Grey boxes represent bacterial families with no identified Ig-coating. The bar plot in the middle panel summarizes the relative distribution of coating types for each of the bacterial families. The dot plot in the upper panel displays the average relative abundance of each bacterial family in bulk feces. (**a**, **b**, **c**, **d**), 'IgA coated' indicates that bacteria are found in the sorted IgA-coated fraction from IgA+ subjects (unique for IgA+), but not in the sorted IgM−coated fraction from IgA− subjects, and vice versa for 'IgM coated' (unique for IgA−); 'IgA+/IgM+' indicates that bacteria are found to be coated with IgA in IgA+ subjects and with IgM in IgA− subjects (shared). 'Strictly non-coated' indicates that bacteria were only identified in non-coated fractions.

none of the coated bacteria to be present in all subjects, which is not surprising based on the known heterogeneous nature of the gut microbiota across individuals[3].

The majority of the coated bacteria belongs to the Firmicutes phylum (*n* = 1532, 93% of coated bacteria, Fig. 3c), dominated by the *Lachnospiraceae* (*n* = 845), *Ruminococcaceae* (*n* = 413) and *Peptostreptococcaceae* (*n* = 83) families that together represented up to 81% of coated bacteria (inner and middle ring, Fig. 3b). Although the average number of *Peptostreptococcaceae* in bulk feces was approximately 60 times lower than that of both *Lachnospiraceae* and *Ruminococcaceae* (Fig. 3d, upper dot panel), the average actual number of coated bacteria of *Lachnospiraceae*, *Ruminococcaceae* and *Peptostreptococcaceae* was similar (Fig. 3d (lower panel displaying the average actual number of coated bacteria per g feces), Supplementary Table 3). The *Peptostreptococcaceae* family members may therefore hold a higher relative Ig-coating compared to the two other families.

Members of the *Akkermansiaceae* family accounted for only 1.6% of all coated bacterial taxa (*n* = 26, Fig. 3b), but a relatively high number of these taxa was coated with IgA or IgM (an average $3.47 \times 10^8$ bacteria/g feces with IgA in IgA+ and $1.75 \times 10^8$ bacteria/g feces with IgM in IgA− subjects, Fig. 3d (lower panel)), representing an average coating frequency of 44% in IgA+ and 40% in IgA− subjects. Moreover, all unique *Akkermansiaceae* ASVs were identified to be coated in at least one of the subjects (Fig. 3d, middle panel). Another notable finding was that taxa in the *Enterobacteriaceae* family were coated by both IgA and IgM (Fig. 3d, middle and lower panel), with an average coating frequency of 23% in IgA+ and 45% in IgA− subjects, which is in line with previous findings in selective IgA deficiency[6], but contrasting a previous report of IgM's inability to coat *Enterobacteriaceae*[10]. However, we also identified a large proportion of strictly non-coated *Enterobacteriaceae* across all subjects (Fig. 3d, middle panel), which could explain this discrepancy between studies.

Amongst the 1766 gut bacteria that were consistently non-coated, at least half of the bacterial taxa in the families *Eggerthellaceae*, *Rikenellaceae*, *Clostridiales* Family XIII, *Clostridiaceae*, *Peptococcaceae*, and *Desulfovibrionaceae* were, like for *Enterobacteriaceae*, also non-coated across all subjects (Fig. 3d, middle panel). Except for *Clostridiales* Family XIII, these families made up relatively high numbers of gut bacteria (up to $1.9 \times 10^9$ bacteria/g feces (Fig. 3d, upper dot panel)), hence it is noteworthy that substantial proportions of these taxa are consistently non-coated.

### IgG assists IgM in coating of most bacterial families in IgA deficiency

We next aimed to determine the extent of coating of bacterial families with IgG together with IgM in IgA− subjects, and with IgA in IgA+ subjects. This was done by FACS-sorting single IgA as well as double-IgAIgG coated bacteria from IgA+ subjects, and single IgM as well as double-IgMIgG coated bacteria from IgA− subjects. The prevalence of bacterial coating with single IgM and double-IgMIgG in IgA− subjects was found to be similar, while the prevalence of bacterial coating with single IgA was significantly higher than that of double-IgAIgG (Fig. 4a,

b). This implies that IgG coating is more frequently assisting IgM in coating of bacteria in IgA− subjects than is the case for IgG and IgA in IgA+ subjects.

### Enhanced coating of potentially flagella- and hexa-acylated lipopolysaccharide-positive bacteria in IgA deficiency

Next, we coupled the coating frequencies of each bacterial family (Fig. 5a) to their anatomical localization, mucus embedding frequencies and immunostimulatory potential to better understand which factors influence whether a bacterium is coated or not. To define the main location of the differentially coated bacterial families in the human gut, we identified metagenomic species based on the publicly available dataset generated by Zmora et al.[19], and our species catalogue published in Moll et al.[15]. The dataset included shotgun sequenced bacterial DNA extracted from mucus and lumen swaps from different sites along the intestinal tract of healthy adults (*n* = 10) from which we calculated the % of mucus-embedded bacterial families in terminal ileum, cecum and descending colon. Most bacterial families with a high Ig-coating fraction (>30 %) were found to have a higher % mucus-embedding in the cecum or large intestine (*Bacteroidaceae*, *Christensenellaceae*, *Clostridiaceae*, *Clostridiales* Family XIII, *Enterobacteriaceae* and *Akkermansiaceae*), than in the small intestine, which was dominated by mucus-embedded *Actinomycetaceae*, *Bifidobacteriaceae*, *Streptococcaceae*, and *Veillonellaceae* of which only *Bifidobacteriaceae* have a high Ig-coating fraction (Fig. 5a, b).

We characterized the immunostimulatory potential of Ig-coated bacteria by performing in silico analysis of their genetic potential to produce the short-chain fatty acid butyrate and the major bacterial ligands flagellin and lipopolysaccharide (LPS), with the latter two activating the immune system via human Toll-like receptor (TLR)5 and TLR4, respectively. Among immunostimulatory bacterial ligands, especially flagellin has previously been shown to be a strong inducer of intestinal IgA responses[20]. Some of the bacterial families did not hold any bacteria with these potentials, and within those that did hold the potential, not all bacteria were identified to carry all necessary genes (Fig. 5c, upper triangle). Amongst those with potential for flagellin production, we observed 0.2–34.5 % Ig-coating of bacteria (Fig. 5c, lower triangle). Genes encoding the enzymes required for production of either penta-acylated or hexa-acylated LPS were found in all the Gram-negative bacteria, as well as in *Veilonellaceae*, as expected[21] (Fig. 5c, upper triangle). *Enterobacteriaceae*, which also possesses flagellin, was the only family amongst which some bacterial taxa also carried genes for production of the pro-inflammatory hexa-acylated LPS variant, while the genes that encode enzymes to produce the penta-acylated LPS variant that does not cause activation of TLR4 in humans[22], were found in eight bacterial families, including the highly abundant *Bacteroidaceae*. Bacteria within five of these families had a high % of Ig-coating (above 12.3 %), while the other three families with potential for penta-acylated LPS production showed limited Ig coating, implying that penta-acylated LPS alone is not sufficient to induce Ig production.

Butyrate promotes increased production of IgA in the small and large intestine through the upregulation of tumor growth factor

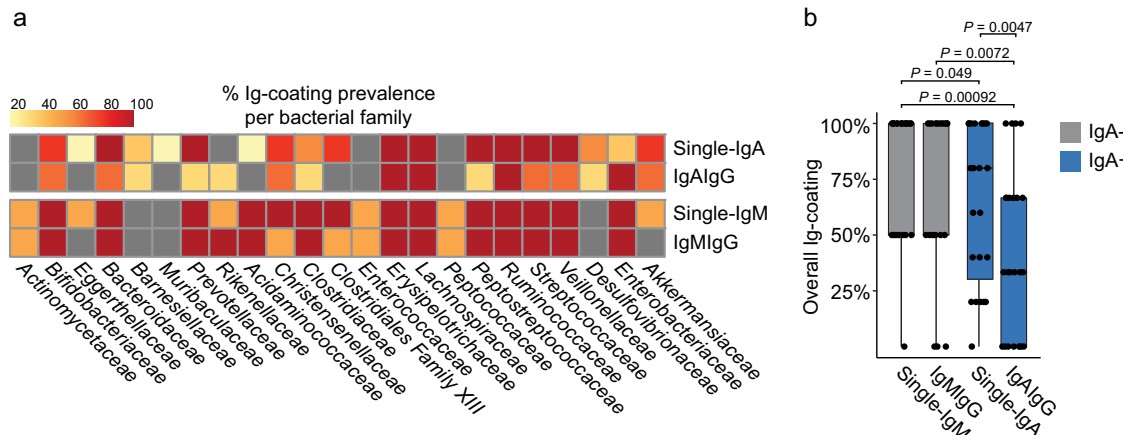

**Fig. 4 | IgG assists IgM in coating of bacteria in the absence of IgA. a** Heatmap showing % prevalence of coated bacterial families within FACS sorted single-IgA and double-IgAIgG coated bacteria from IgA+ subjects, and single-IgM and double-IgMIgG coated bacteria from IgA- subjects. Bacteria were identified upon 16 S rRNA gene amplicon sequencing identifying uniquely coated bacteria based on amplicon sequence variants (ASVs). Grey boxes represent families with no identified Ig-coating. **b** Overall prevalence of Ig-coated bacteria for each Ig combination from (**a**) (n = 23). Wilcoxon test was used to compare the coating prevalence amongst all Ig combinations. *P* values are two-sided and adjusted using FDR. A grey fill indicates IgA− subjects, while a blue fill indicates IgA+ subjects. For boxplots, the center line indicates the median and the box limits indicate the quartiles. Whiskers extend to the data points within 1.58× the interquartile range, and outliers are shown as individual dots.

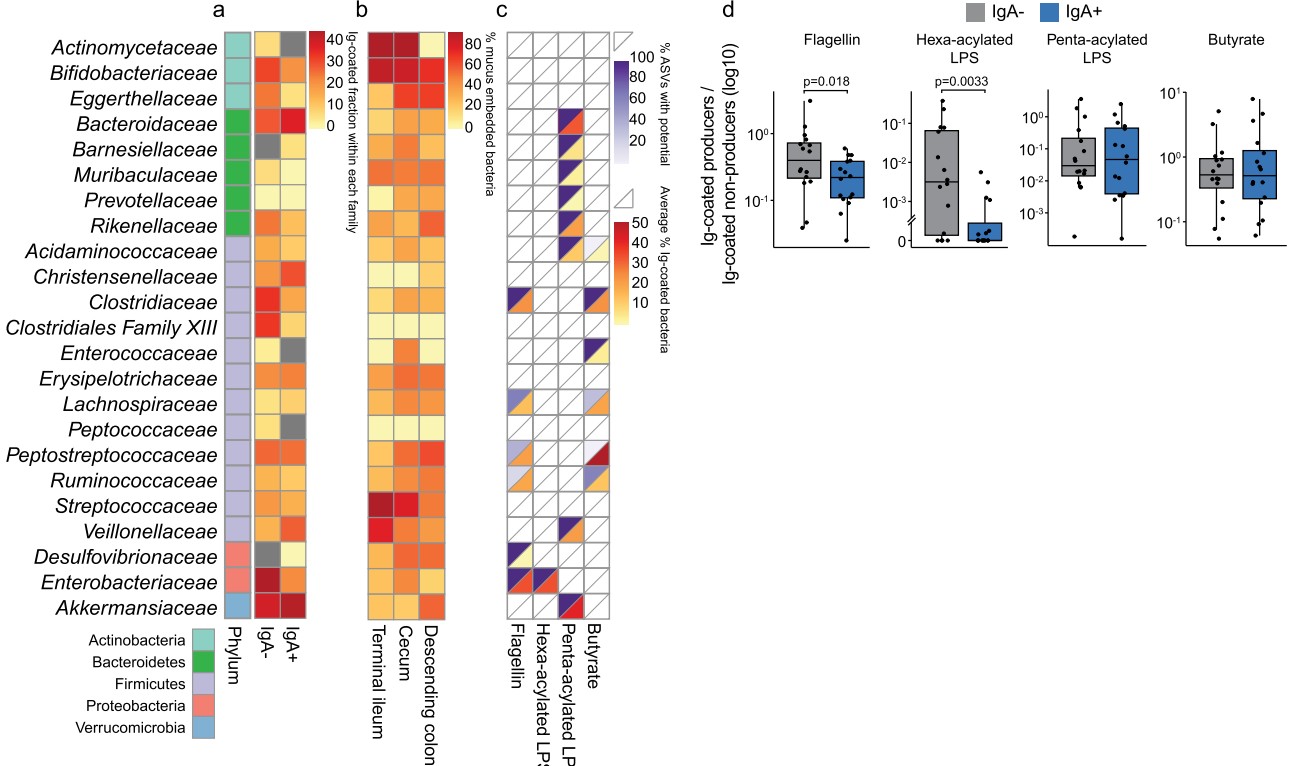

**Fig. 5 | Enhanced coating of potentially flagella- and hexa-acylated lipopolysaccharide-positive bacteria in IgA deficiency. a** The average Ig-coated fraction per g feces within each bacterial family (load of Ig-coated vs. total load of bacterial family) from paired samples of IgA+ and IgA− subjects (n = 2 × 16). Grey boxes represent families with no identified Ig-coating. **b** Heatmap showing the % of bacteria embedded in mucus compared to their mean abundance in the lumen at three different sites along the intestine in healthy adults (n = 10), Zmora et al.[19]. **c** The upper triangle represents % of ASVs predicted to hold the immunostimulatory potential, while the lower triangle represents the average frequency of Ig−coating of bacteria with the immunostimulatory potential (load of Ig−coated vs. total load of bacteria with immunostimulatory potential, n = 2 × 16). **d** Boxplots showing the ratio between load of Ig-coated bacteria with the immunostimulatory potential vs. load of Ig-coated bacteria without the immunostimulatory potential in IgA− (n = 16) and IgA+ (n = 16) subjects. Ig-coating data derive from sorted and sequenced Ig-coated bacteria from Fig. 3. Paired Wilcoxon test was used to compare IgA− and IgA + pairs. *P* values are two-sided and not adjusted for multiple comparisons. A grey fill indicates IgA− subjects, while a blue fill indicates IgA+ subjects. For boxplots, the center line indicates the median and the box limits indicate the quartiles. Whiskers extend to the data points within 1.58× the interquartile range, and outliers are shown as individual dots.

(TGF)-β production[23]. However, we found varying Ig-coating of the six bacterial families with potential for butyrate production (Fig. 5c), although these families are found to reside in the mucus layer of cecum or the colon (Fig. 5b). Notably, more non-producers than producers were found to be Ig-coated, as the ratio between the numbers of coated bacteria was below 1 for each of the ligands (Fig. 5d). However, IgA− subjects displayed enhanced Ig-coating of bacteria predicted to be flagellin and hexa-acylated LPS producers.

Amongst the predicted hexa-acylated LPS producing species, we identified *Escherichia coli* as the most abundant. As some *E. coli* spp. are gut pathobionts that induce inflammation if they are uncontained

in the host, we went on to examine the Ig-coating patterns of *E. coli* in more detail.

### IgG subtypes assist IgM in coating of *E. coli* in IgA deficiency

We found *E. coli* numbers to be 7.4-fold higher in the 31 household pairs of IgA− as compared to IgA+ subjects in the present study (Fig. 6a, $P = 0.012$). This finding is reported earlier in a larger subgroup of the same cohort[15]. The fraction of *E. coli* that was coated with any IgM (single or double) in IgA− subjects is significantly higher than the fraction of any IgA coated *E. coli* in IgA+ subjects (median of 30% [0.61%; 86%] vs. median of 10% [0%; 36%], $P = 0.009$, Fig. 6b). The

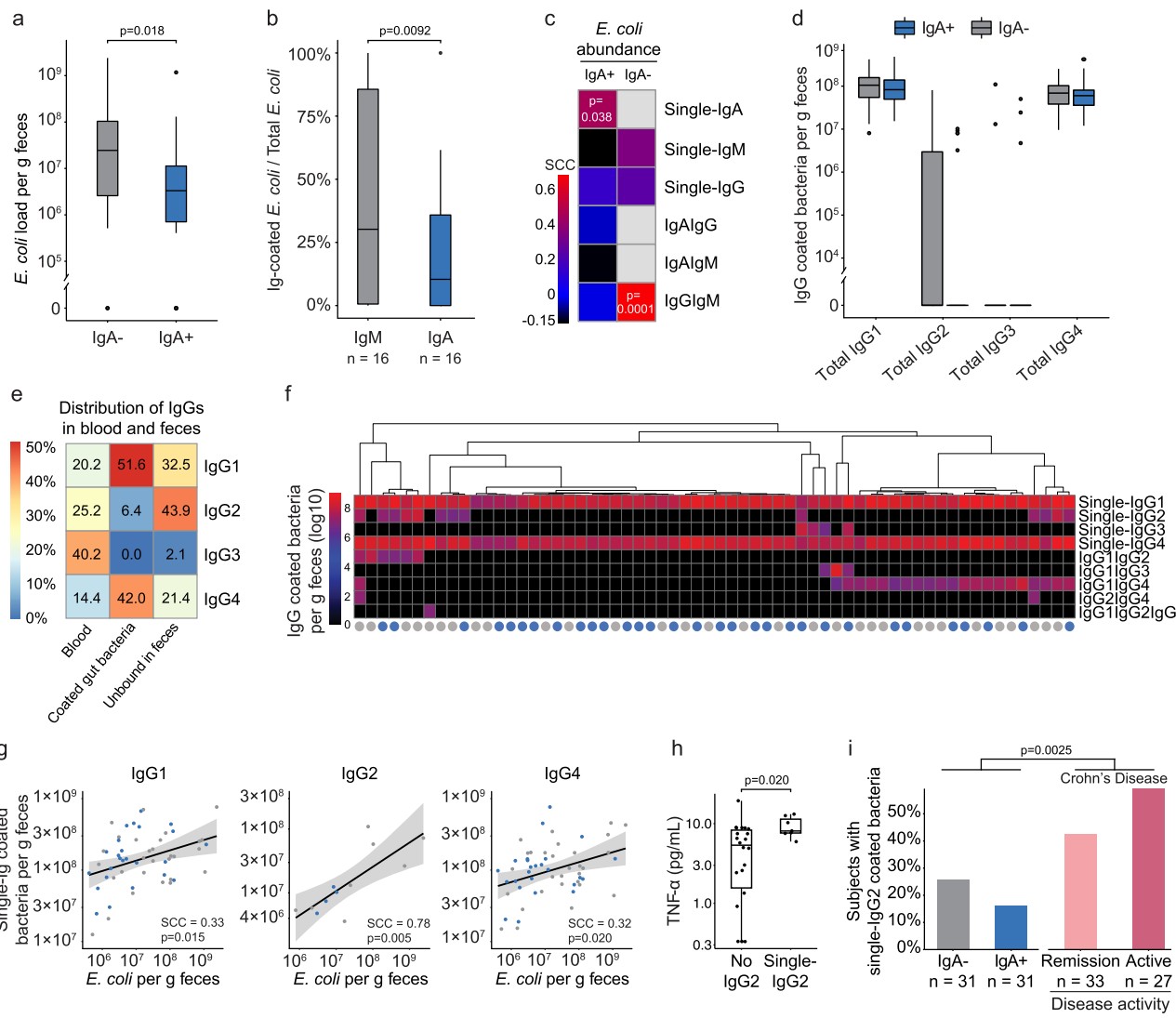

**Fig. 6 | IgG assists IgM in coating of *E. coli* in the absence of IgA. a** Load of *E. coli* per g feces in IgA+ and IgA− subjects ($n = 2 \times 31$). **b** Fraction of *E. coli* coated with total IgA or IgM in IgA+ or IgA− subjects (based on sorted and sequenced data from Fig. 3, $n = 2 \times 16$ pairs). **c** Spearman rank correlation coefficient (SCC) between *E. coli* load and Ig-coating in IgA+ and IgA− subjects. The colored bar indicates SCC, while light grey indicates absence of coating ($n = 2 \times 31$). **d** Number of gut bacteria coated with IgG1, IgG2, IgG3 and IgG4 in IgA+ and IgA− subjects ($n = 2 \times 31$). **e** Distribution of IgG1–4 in blood, on coated bacteria and in fecal water as non-bacterium bound. Determined by ELISA from 14 cohort subjects with simultaneously collected blood and feces. **f** Complete-linkage clustering of single- and multi-IgG1–4 coated bacteria, represented as coated bacteria/g feces ($n = 2 \times 31$). Blue: IgA+ subjects, Grey: IgA− subjects. **g** Correlation (SCC) between load of *E. coli* and bacteria coated with single-IgG1 ($n = 54$), single-IgG2 ($n = 12$), and single-IgG4 ($n = 54$). The straight lines and the shaded areas indicate the parametric correlation

coefficient and the 95% confidence interval, respectively. *P* values are two-sided and not adjusted for multiple comparisons. **h** Systemic TNF-α levels in a subgroup of subjects with ($n = 7$) and without ($n = 22$) single-IgG2 coated bacteria. **i** Prevalence of subjects with single-IgG2 coating in Crohn's disease (CD) patients during active disease or remission compared to IgA− and IgA+ subjects. Differences in single-IgG2 prevalence between CD and the IgA cohort were tested using Fisher's exact test. The *P* value is two-sided. A grey fill indicates IgA− subjects, while a blue fill indicates IgA+ subjects. **a–d**, paired Wilcoxon test was used to compare IgA− and IgA+ pairs. *P* values are two-sided and adjusted using FDR, and only significant *P* values are shown. **h** Unpaired two-sided Wilcoxon test was used to compare single-IgG2 to no IgG2 subjects. For boxplots, the center line indicates the median and the box limits indicate the quartiles. Whiskers extend to the data points within 1.58× the interquartile range, and outliers are shown as individual dots.

fraction of IgM coated *E. coli* did however not correlate with the total load of *E. coli* (Supplementary Fig. 3, $P = 0.15$), indicating that IgM coating of *E. coli* might be insufficient to fully control its growth. We next examined relationships between single- and double-Ig coating and the *E. coli* load, and found *E. coli* load to positively correlate with single IgA coating in IgA+ subjects (SCC = 0.46, $P = 8.47 \times 10^{-3}$, Fig. 6c), but not with double-IgAIgG coating (Supplementary Table 4). In IgA− subjects, the *E. coli* load correlated strongly with double-IgGIgM coating (SCC = 0.69, $P = 1.59 \times 10^{-5}$), but not with single IgM. Combined, this indicates that IgA coating is sufficient in controlling *E. coli* numbers, while in the absence of IgA, IgM alone may not suffice, and the assistance of IgG may be needed.

The findings regarding IgG in IgA− subjects motivated us to further investigate bacterial coating by the IgG subtypes: IgG1, IgG2, IgG3, and IgG4. When examining the overall bacterial coating with IgG1 to IgG4, it was apparent that there were major differences in the number of bacteria coated by each subtype (Fig. 6d, e). IgG1 and IgG4 showed by far the highest coating-degree, with an average coating of nearly $10^8$ bacteria/g feces for each subtype (Fig. 6d, Supplementary Table 5). Due to the observed extensive IgG1 and IgG4 coating of bacteria, we made a subanalysis by ELISA using samples from 14 study subjects with simultaneous blood and fecal collection to examine IgG1, IgG2, IgG3 and IgG4 gut bacterial coating ratios in relation to those of circulating blood and bacteria-free fecal water. From this, it was apparent that IgG1 and IgG4 predominantly bound to gut bacteria, while IgG2 was present in the gut lumen, but not bound in high frequencies to bacteria (Fig. 6e). Although IgG2 has long been recognized to be induced in response to microbial pathogens possessing repeating carbohydrate antigens (encapsulated microbes), its possible role in binding to non-bacterial members, such as viruses, within the human intestinal tract has not been addressed thoroughly. An earlier study identified that terminal ileum IgG+ plasmablasts (IgG1: 51.8%, IgG2: 35.9%, IgG3: 2.5% and IgG4: 9.9%) react largely antigen-specific to a panel of commensal and pathogenic gut bacteria, and virus-specific binding was also detected[24]. A specific role for IgG2 in virus-binding was however not specified.

When examining the patterns of bacterial coating, it appeared that IgG1 and IgG4 displayed similar and consistent coating percentages across IgA+ and IgA− subjects (Fig. 6f), which could indicate that IgG1 and IgG4 take part in a normal barrier-maintaining response during homeostasis. However, IgG1 and IgG4 did not consistently coat the same bacteria across all subjects, as only some subjects showed bacterial double-coating with IgG1 and IgG4 (IgA−: 45%, IgA+: 26%, Fig. 6f). Moreover, we observed that single IgG2- and IgG3-coated bacteria were only identified in relatively few subjects (IgG2: $n = 13$, IgG3: $n = 4$), and always with a low coating percentage relative to IgG1 and IgG4.

Abundances of IgG1, IgG2 and IgG4 single-coated gut bacteria correlated with a SCC of 0.33, 0.78, and 0.32, respectively to the total fecal *E. coli* load across all subjects (Fig. 6g, Supplementary Table 4). For IgG1 and IgG4 this was comparatively similar to the SCC of 0.46 seen for single-IgA coating vs. *E. coli* load (Supplementary Table 4). This suggests that bacterial coating with IgG1 and IgG4 might play a role as barrier-maintaining factors in healthy subjects. In subjects displaying single-IgG2 bacterial coating (21% of subjects), we found higher systemic levels of the pro-inflammatory cytokine TNF-α than in subjects with no single-IgG2 bacterial coating (Fig. 5h, $P > 0.01$). Although single-IgG2 bacterial coating was relatively sparse, we found it to be 1.6-fold more prevalent in IgA− vs. IgA+ subjects, whilst prevalence increases by an additional 2-fold in Crohn's disease patients (Fig. 5i). This points to single-IgG2 bacterial coating being enhanced during inflammatory processes.

## Discussion

We here demonstrated that IgG assists IgM in coating of bacteria in IgA deficiency. IgG was found to coat less bacteria than IgA, but still a

sizable fraction (27% of IgA coated), indicating that IgG coating of bacteria is significant in healthy individuals and may play a role in controlling gut bacteria. Amongst the IgG subtypes, especially IgG1 and IgG4 bacterial coating were found to dominate. Up to on average 7% of fecal bacteria were coated with one or more antibodies, and 36% of these overlapped in IgA+ (any IgA coating) and IgA− (any IgM coating) subjects. Approximately 32% of the fecal bacteria were single or multi-coated with IgA, but not with IgM in the absence of IgA. The majority of the bacteria coated with single- or multi-IgA but not by single- or multi-IgM were Gram-positive, primarily of the Firmicutes phylum, in line with previous studies[8]. The same proportion of bacteria (32%) were exclusively coated with single- or multi-IgM in the absence of IgA. A few previous studies in humans have identified IgG bacterial coating mounting to the same level as in the current study (median: 1.7%) (adults[16], infants[17]), while another study identified minute levels of bacterial IgG coating in humans (median: 0.03%)[13]. A difference in the method used for fecal pellet preparation, where Fadlallah et al.[13]. employed a method optimized for bacterial proteomics is speculated to explain the discrepancy.

The % IgA-coating was previously shown to be high in the small intestine and decrease along the intestinal tract[2]. In our study we found % IgA coating to inversely associate with numbers of bacteria per gram of feces, indicating that IgA may be produced in a continuous manner independently on the number of bacteria at the site. This finding is supported by previous studies on differential bacterial load along the intestinal tract, showing increasing bacterial load in the colon[25], implying that fecal consistency and bacterial load should be taken into consideration when comparing studies.

It has been demonstrated that IgA plays an important role in blocking mucosal uptake of microbial antigens, especially flagellin[20], a ligand proposed as an important driver of Crohn's disease via TLR5[26]. According to the present in silico analysis, members of both the *Clostridiaceae*, *Lachnospiraceae Peptostreptococcaceae Ruminococcaceae Desulfovibrionaceae*, and *Enterobacteriaceae*, families hold the potential to synthesize flagellin, but they appeared to differ considerably in coating levels, and potential for flagellin production did not result in preferentially higher coating as compared to coating frequencies in bacteria without this potential. Members of *Akkermansiaceae*, despite being highly coated, were found not to carry flagellin, nor do they activate TLR5[27]. Rather, they are known to be mucus-degrading and embedded in the mucus layer[28]. The coating frequency of embedded bacteria would expectedly be higher for bacteria located in the colonic mucus[29], as secreted IgA is concentrated within mucus[30]. This is also consistent with the present finding of a lower coating frequency of small intestinal mucus-embedded bacteria.

Based on metagenomics sequencing, we identified fecal *Enterobacteriaceae* in this cohort to consist mainly of *E. coli* species, which is in accordance with a recent study of the fecal microbiota of healthy individuals[31]. Since we identified significant enrichment of *E. coli* in IgA-deficient subjects, while being largely single- or multi-IgM coated, it is an open question if IgM is equally effective as IgA in delimiting growth of *E. coli*. On the other hand, since subjects with IgA deficiency in general show few signs of acute illness or infection[32], this suggests that secreted antibodies generally bind to antigens used by *E. coli* spp. for possible invasion of epithelial cells. This speculation was further substantiated in a recent study, where we used strain variation analysis of *E. coli* and identified that various strains of gut *E. coli* in asymptomatic subjects hold many virulence factors used for gut epithelial translocation[15]. It is therefore likely that human beings are able to control *E. coli* invasiveness during asymptomatic phases by Ig-coating of *E. coli*. While single-IgA coating was the only Ig-coating type found to correlate with the fecal load of *E. coli* in IgA+ subjects, we found bacterial single- and double-coating with both IgM and IgG to dominate in IgA− subjects. The latter also correlated positively with *E. coli* load. Thus, in a complementary manner, IgG and IgM may compensate for the lack of IgA,

although the increased load of *E. coli* in feces of IgA-deficient subjects indicates that growth of *E. coli* is handled less effectively by IgM and IgG coating as compared to IgA coating in IgA+ subjects, while invasiveness may be fairly controlled, as IgA− subjects experience few symptoms[4]. However, we find a 1.6-fold increased prevalence of IgA− vs. IgA+ subjects with single-IgG2 bacterial coating. Factors promoting isotype switching to IgG2 are largely unexplored[33], but it is reported that IgG2 may be induced by the type 1 immune cytokines IFN-γ, IL-12 and IL-18[34,35], by specific pathogenic microbes[35,36], and in response to *E. coli*-derived LPS stimulation in vitro[37], thereby supporting a link between the presence of hexa-acylated LPS-producing bacteria, such as *E. coli*, and IgG2 production. Enhanced systemic TNF-α levels in subjects with single-IgG2 coated bacteria, and increased prevalence in Crohn's disease patients with active disease, further substantiate that single-IgG2 coating is linked to local and systemic inflammation, and may be a marker for on-going inflammatory processes.

Our finding of a universal, and predominant bacterial coating with IgG1 and IgG4 across all subjects is interesting, and has not been reported before. Following induction by IL-4[38] or IL-13[39], IgG4 class-switched B cells may enhance IgG4 responses by optimizing their differentiation into IgG4-secreting plasma cells in response to IL-10 and TGF-β[40–43], two cytokines also required for IgA class switching and IgA production. IL-10 and TGF-β are regarded as anti-inflammatory cytokines produced during homeostatic conditions in the gut. IgG4 might act as a mediator for tolerance via its low affinity binding to activating compared to inhibitory Fcγ receptors and its inability to initiate the complement cascade[44]. In addition, IgG4 is able to perform Fab arm exchange in vivo[45], which combine specificities of two IgG4 molecules effectively preventing crosslinking of antigens. As it was not the main focus of the current study to elucidate mechanisms of IgG1 and IgG4, we can only speculate that one or both of them may be involved in processes similar to that of IgA, hence assisting in maintaining an inner mucus layer relatively free of microorganisms without giving rise to excessive immune reactions towards the commensal members of the microbiota[46].

In summary, we have established a role for IgG coating of gut bacteria, which is dominated by the IgG1 and IgG4 subtypes. This coating was uniformly distributed across all subjects and independent of IgA status. IgA-deficient subjects displayed 1.6-fold higher prevalence of single-IgG2 bacterial coating that linked to fecal *E. coli* load and concomitantly increased TNF-α levels in blood. Single-IgG2 coating was even more prevalent in Crohn's disease patients, hence suggesting a role for single-IgG2 coating during inflammatory conditions. In the presence of IgA, single-coating with IgA was mostly prominent, while, single- and double-coating of bacteria with IgM and IgG dominated in the absence of IgA. This switch to bacterial double-coating in IgA deficiency could be one of the reasons for the partial bacterial-specific barrier defense experienced in IgA-deficient humans, and explain why they experience relatively mild disease symptoms over a life time[4].

## Methods
### Cohort design
The study was approved by the regional ethical committee, Stockholm, Sweden (No: 2016/2502-31/2). Fecal and blood samples from patients with IgA deficiency (<0.07 g/L of serum IgA) and household members were collected at the Department of Clinical Immunology at the Karolinska University Hospital, Huddinge, Sweden, and blood samples were analysed for IgA, IgG, IgM and cytokines as detailed in Moll et al.[15]. All participants provided written informed consent. The current study is based on samples from 31 out of 50 household pairs for which sufficient material was available to conduct Ig-coating analyses. None of the subjects reported any antibiotic treatment for at least 60 days before sampling, nor any ongoing gastrointestinal problems, including inflammatory bowel disease, celiac disease or lactose intolerance.

Samples for comparison to single-IgG2 coating prevalence in Crohn's disease patients were derived from the IBD South Limburg (IBDSL) cohort, which is a population-based inception cohort from the South Limburg area of the Netherlands, enrolled upon written informed consent, collected as described[47], approved by the local Medical Ethics Committee, and registered in http://www.clinicaltrials.gov (NCT02130349).

### Determination of fecal bacterial load and Ig-coating by flow cytometry
Fecal samples were incubated on ice for 1 h in PBS at 100 mg/mL. The samples were homogenized, spun down (15 min, 50 g, 4 °C), and the supernatant was aspirated. Supernatant was centrifuged (5 min, 8000 g, 4 °C), washed twice and resuspended in buffer 1 (PBS + 1% BSA (>98% Sigma-Aldrich)). An aliquot was diluted 150× in buffer 2 (PBS + 1% BSA + 0.01% Tween20 + 1 mM EDTA), mixed with DAPI to a final concentration of 1 mM and counting beads (BD Biosciences), and counted on a FACSCanto II flow cytometer (BD Biosciences). The bacterial population and counting beads were gated based on SSC-A/Pacific-Blue. Determination of bacterial counts was performed based on the manufacturer's protocol. A volume of sample equivalent to a total of $150 \times 10^6$ bacteria was then transferred to a new tube for the Ig-coating analysis and FACS sorting. Bacteria were incubated for 20 min at 4 °C in buffer 1 containing 20% mouse serum (Sigma-Aldrich). The sample was stained in a mixture of 1:10 Anti-Human IgA-PE (Miltenyi Biotec, clone IS11-8E10), 1:20 IgM-APC/Cy7 (Biolegend, clone: MHM-88), 1:10 IgG-PE/Cy7 (Biolegend, clone: HP6017) and buffer 1 for 30 min at 4 °C. The sample was washed twice in buffer 1 and resuspended in buffer 2. An aliquot was diluted and incubated with DAPI before being analyzed on a FACSCanto II flow cytometer (BD Biosciences). The procedure for analyzing the differential coating with IgG subtypes was performed as described above, but with the following antibodies: 1:40 Anti-Human IgG1-Biotin (clone: HP6001), 1:20 Streptavidin-APC-Cy7, 1:40 Anti-Human IgG2-AF647 (clone: HP6002), 1:40 Anti-Human IgG3-AF488 (clone: HP6050) and 1:4 Anti-Human IgG4-PE (clone: HP6025) (all SouthernBiotech). The analyses were based on 50,000 recorded events and gated on DAPI⁺ cells. Fluorescence-minus-one (FMO) control staining's were conducted for all Igs to ensure proper gating setup (Supplementary Figs. 1 and 2).

### Sorting and sequencing of Ig-coated and non-coated fecal bacteria
The total IgM⁺ population (including single and double-coated bacteria) was sorted from 16 IgA deficient subjects, and the total IgA⁺ population (including single-, double- and triple-coated bacteria) from the corresponding IgA+ household member. For a subgroup of subjects we also sorted specifically single-IgA, -IgM and -IgG in addition to double-IgAIgG and double-IgMIgG coated bacteria. From all subjects, we also sorted out non-coated bacteria (triple-negative for IgA, IgM and IgG). We collected at least $2 \times 10^6$ coated bacteria or non-coated bacteria using a MoFlo XDP cell sorter (Beckman Coulter). Cells were pelleted upon sorting (5 min, 8000 g, 4 °C) and stored at −80 °C until DNA extraction. Details on sequencing methodology are available in Supplementary materials and methods. A total of 1652 coated and 1766 strictly non-coated bacteria, determined as unique amplicon sequence variants (ASVs), were identified across all samples.

### Reconstruction of the phylogenetic relationship between coated and non-coated fecal bacteria
The ASV sequences were used to reconstruct the phylogenetic relationship between coated and non-coated fecal bacterial species. Non-coated fecal bacteria were defined as ASVs which were never found in

any of the coated samples but only amongst non-coated fecal bacteria sorted samples. The nucleotide sequences were aligned using MAFFT v. 7.4.53 using standard settings. An approximately-maximum-likelihood phylogenetic tree was inferred using FastTree v. 2.1.10, and the Jukes-Cantor substitution model and a gamma model with 20 rate categories. The phylogenetic tree was visualized as a midpoint root tree using R v3.5.1.

### In silico based inference of immunostimulatory ligands and butyrate production capability amongst coated bacteria

Metagenomic species were annotated for their capacity to produce hexa- or penta-acylated LPS, flagellin and butyrate using KEGG Modules v. 2. Each gene in the IGC gene catalog were assigned KEGG orthology groups (K numbers v. 2 (KOs)) by using GhostKOALA v. 2.1 (https://www.kegg.jp/ghostkoala). For each species, the flagellin production was evaluated by the presence of K numbers responsible for the production of flagellin as described in the flagellar assembly map (map02040). The type of LPS, or whether a bacterium was capable of producing it at all, was evaluated by each species ability to convert UDP-N-acetylglucosamine to $KDO_2$-lipid A as described in the lipopolysaccharide biosynthesis pathway (map00540)[21]. The butyrate production capacity was evaluated based on the presence of individual K numbers responsible for converting succinate semialdehyde, acetoacetyl-CoA, vinylacetyl-CoA, glutaconyl-1-CoA or butanal to butanoyl-CoA, which can finally be converted to butyrate as described in the butanoate metabolism map (map00650).

### ELISA determination of IgG subtypes

MaxiSorp 96-well plates (Thermo Fisher) were coated overnight with Anti-human IgG1, IgG2, IgG3, or IgG4 capture antibody (clone: HP6001, HP6002, HP6050, HP6025, respectively, all SouthernBiotech) in carbonate buffer (pH 9.6) at 4 °C at a concentration of 5 μg/mL. Plates were then washed four times in washing buffer (PBS + 0.05% Tween20), blocked in PBS + 1% BSA for two hours at room temperature with shaking, and again washed four times in washing buffer. Diluted samples thawed on ice, diluted standard (polyclonal human IgG, SouthernBiotech), and negative control (PBS + 1% BSA) were added in duplicates. Fecal water samples (aspirated supernatant after centrifuging homogenized fecal samples (100 mg/mL in PBS)) were diluted 2× in PBS + 1% BSA, while blood samples were diluted 120,000 times for IgG1 detection, 70,000 times for IgG2 detection, and 20,000 times for IgG3 and IgG4 detection. Plates were incubated at room temperature for two hours with shaking, washed four times in washing buffer, and incubated with Anti-Human IgG HRP-conjugated detection antibody (SouthernBiotech) for one hour at room temperature with shaking. Plates were then washed four times in washing buffer and one time in demineralized water before incubation in substrate solution (TMB in peroxidase) for 15 min at room temperature in darkness. The reaction was stopped with 2 M phosphoric acid and absorbance measured at 450 nm (630 nm reference) on a microtiter plate reader (PowerWave$_x$ Bio-tek Instruments). Concentrations of IgG1–4 were determined based on the standard curves.

### Statistics and reproducibility

The study sample was based on fecal samples collected from 31 out of 50 household pairs for which sufficient material was available to conduct Ig-coating analyses. No statistical method was used to predetermine this sample size. R v3.5.1 was used for statistical analyses and data visualization. Flow cytometry data were analyzed using FlowJo (Version 10.5.0, Tree Star, Inc., Ashland, OR) and gated according to Supplementary Fig. 1 and 2. All data were analyzed as household pairs, and randomized during laboratory processing. The paired setup prevented the investigators from being fully blinded, but otherwise all analyses were conducted in a blinded manner. No data

were excluded from the analyses. Nonparametric tests were used on all data: Wilcoxon rank-sum test was used when comparing two groups and the Spearman's rank correlation coefficient (SCC) to define correlations. Paired Wilcoxon rank-sum test statistics were used when comparing IgA deficient subjects with IgA sufficient household members. For calculation of the ratio of the quantitative abundance amongst total IgA- or IgM-sorted bacteria, zeroes were replaced with 1, and 1 was also added to the remaining data. When relevant, $P$ values were adjusted for multiple testing using false discovery rate (FDR), and considered statistically significant when $P < 0.05$ and $q < 0.1$.

### Reporting summary

Further information on research design is available in the Nature Portfolio Reporting Summary linked to this article.

## Data availability

Source data are provided with this paper. The raw nucleotide datasets have been deposited in the European Nucleotide Archive repository under accession number: PRJEB55323 or National Center for Biotechnology Information Sequence Read Archive (SRA) under accession number: PRJNA633381. Sequence data used to identify mucus-embedded bacteria are deposited in the European Nucleotide Archive under accession number: PRJEB28097. The fastq-files from the FACS-sorted samples are available upon request. Assignment of taxonomy was done using the SILVA database v. 132 [https://doi.org/10.5281/zenodo.1172783]. Other non-GDPR restricted data generated during and/or analyzed during the current study are available from the corresponding author upon request. Individual-level personally identifiable data from the individuals participating in the cohort cannot be made freely available, to protect the privacy of the participants, in accordance with the Danish Data Protection Act and European Regulation 2016/679 of the European Parliament and of the Council (GDPR) that prohibit distribution even in pseudo-anonymized form. However, research collaborations are welcome, and data can be made available under a joint research collaboration by contacting Susanne Brix (sbrix@dtu.dk). Source data are provided with this paper.

## Code availability

No custom algorithms were created as part of this study. See methods section for details of software used in the analysis.

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

## Acknowledgements

We thank Renée Engqvist at Karolinska Instituttet for help during sample collection. This research was supported by Technical University of Denmark providing a PhD scholarship to CE in the group of SB, the Swedish Research Council (2019-0132, QPH), and the Knut and Alice Wallenberg Foundation (KAW 2020-0102, LH and QPH). SB is the incumbent of the FII institute Research Chair at DTU in Immune-based Prediction of Disease.

## Author contributions

Conceptual design of the study and method implementation: C.E., S.B. Responsible for IgA deficiency cohort: L.H. and Q.P.H. Generation of Ig-

coating profiles: C.E., A.R.A.P. Data analysis: C.E. supported by J.M.M., P.N.M., S.B. Sorting of coated bacteria: C.E., L.B.R. supported by S.B. Implementation of volume-based method and sequencing of coated bacteria: C.E. supported by N.B.D.S., M.D.D., K.K., S.B. ELISA: L.B.R., supported by CE, RID and SB. Responsible for Crohn's disease cohort: J.P., DMAEJ. Writing of the manuscript: C.E., S.B. Critical revision of manuscript: all authors.

## Competing interests

The authors declare no competing interests.
