## [Peer Review File · Nature Communications]

IgG and IgM cooperate in coating of intestinal bacteria in IgA deficiencyREVIEWER COMMENTS

Reviewer #1 (Remarks to the Author):

In this manuscript, Eriksen C. et al. show that IgG recognized fecal bacteria together with IgM when IgA was absent. IgG1 and IgG4 subclasses dominated bacteria-coating IgG antibodies independently of the IgA status. Of note, a significant fraction of fecal bacteria from a subset of IgA-deficient patients were coated by the IgG2 subclass. These patients also displayed increased circulating IgG and TNF-alpha, two hallmarks of inflammation possibly associated with a compromised gut barrier.

GENERAL COMMENT

This is a well-written manuscript that explores the impact of human IgM and IgG antibodies, including IgG subclasses, on the coating of fecal bacteria in individuals that lack IgA (i.e., in patients with IgA deficiency). Remarkably, IgA-sufficient household were used as controls, which is considered an important strength of this study. Elegantly presented data from well-performed experiments support the main conclusions. The statistical analysis of these data is considered appropriate. Weaknesses include the descriptive nature, poor mechanistic insight and limited novelty of this study. In particular, the novelty of this work is greatly attenuated by previously published reports showing binding of gut commensals by IgG and/or IgM in IgA-deficient patients and IgA-sufficient donors (refs. 6, 8, 10, 13, 17). The following additional comments are provided to enhance the impact of this manuscript.

SPECIFIC COMMENTS

- 1) "Gut microbiota" should be more accurately defined as "fecal microbiota". One key difference between fecal and gut commensal bacteria relates to the very low percentage of fecal microbes coated by secretory IgA (~ 7-10%) compared to gut microbes (~ 70-90%). The manuscript does not discuss this crucial difference and its potential implications.
- 2) Please, revisit the following sentence, which seems incomplete/inaccurate. "The findings regarding IgG in IgA- subjects motivated us to further investigate coating by the IgG subtypes: IgG1, IgG2, IgG3, and IgG4, which make up for the total IgG levels in the previous."
- 3) The increase of commensal-reactive IgG2 in a subset of IgA deficient patients is quite interesting, but should be further discussed in the context of earlier data by Benckert J. et al. (J Clin Invest, 2011). In this published study, the gut mucosa from IgA-sufficient individuals includes antigen-specific plasma cells expressing IgG1, IgG2 or IgG4 in addition to IgM, IgA1 or IgA2.
- 5) Authors claim that "IgG4 is believed to be induced by the same isotype switching cytokines as IgA, namely IL-10 and TGF-beta41," This is inaccurate. The only universally known cytokines capable of inducing IgG4 are IL-4 and, to some extent, IL-13 (reviewed in PMID: 15229473; PMID: 18370922, PMID: 22728528; PMID: 25411432, etc.). In this regard, ref. 41 shows no data in support of the IgG4 class switch-inducing function and, more in general, the IgG4-inducing function of IL-10 and TGF-beta. Nonetheless, the involvement of these anti-inflammatory cytokines in the induction of IgG4 is plausible, particularly in the context of a "regulatory brand" of Th2 responses. However, also these responses would almost certainly require B cell-activating STAT6-inducing signals from IL-4 and/or IL-13 for the induction of IgG4.
- 6) The homeostatic function of IgG4 in the gut mucosa may be independent of non-inflammatory environmental signals (i.e., IL-10 and TGF-beta) required for its induction and rather be linked to the poorly pro-inflammatory nature of the heavy chain Cgamma4 region of IgG4. Indeed, Cgamma4 does not have any prominent pro-inflammatory effector function, being unable to engage activating Fcgamma receptors or initiate the complement cascade. These crucial biological aspects of IgG4 are not discussed.
- 7) No direct evidence is provided to show the protective activity of IgM and IgG on the mucosal barrier in IgA deficiency. In particular, no direct evidence is provided to show the perturbation of

the intestinal barrier in IgA-deficient patients with increased IgG2-coated bacteria as well as circulating IgG and TNF-alpha.

Reviewer #2 (Remarks to the Author):

In their manuscript "IgG and IgM cooperate in protection of the mucosal barrier in IgA deficiency", Eriksen and coworkers study antibody covering of the gut microbiota in IgA proficient and deficient (IgAD) individuals from the same households. They find that IgM substitute for IgA in IgAD, that IgG is more often covering bacteria than previous studies have suggested, although they do not see differences depending on IgA status. Dual or triple coating by several Ig classes is not uncommon. They compare the bacterial composition of IgM coated species (from IgA deficient individuals) with IgA coated species (from IgA proficient individuals). Based on previously published datasets, they determine the likely localization of antibody coated and non-coated bacteria, and found that presence of flagellin and potential to produce acetylated LPS was associated with IgA coating, while this was less pronounced for butyrate production. E coli numbers increased in IgA deficient individuals, in which E. coli was also more often covered with IgM in deficient individuals than with IgA in proficient ones. IgG1 and IgG4 were the most prevalent classes coating gut bacteria, while IgG2 was present in faces but did not coat bacteria.

In general, this is a relatively well written report of experiments performed on IgA deficient individuals, the largest immunodeficient group. As these in general have a rather mild phenotype despite lacking the antibody class that is produced in largest quantities, it is important to study to which extent compensatory mechanism (such as secretion of other antibody classes) may explain this. Relatively few studies have been presented that study this in depth, and for this reason the study is of interest.

Strengths of the study

The authors study a relatively large cohort of IgA proficient and deficient individuals from the same households to diminish variability due to selection

The authors demonstrate that IgM coating appear to take over the coating role of IgA in the absence of it.

The authors find that IgG coating from IgG1 and IgG4 antibodies is more prevalent than previously thought, but that IgG appear to play a minor role in substituting for IgA.

Weaknesses

Other studies have been published that have studied the composition of IgA bound vs IgM bound in IgA deficient individuals, so the study is not unique in this manner.

Data from individuals from the same households are not used for paired analysis, which may have revealed additional information. In the absence of pairing, it is unclear whether the use of household samples will in fact decrease variability for each group.

It appears to be a follow up study on IgA deficient individuals from the same group (Moll et al. Gastroenterology 7: 2423). The overlap (if any) between the cohorts in previous and this study is not clear. Some of the findings (i.e. increased levels of E. Coli in IgA deficient individuals) were reported in the previous study.

It is unclear what the authors try to achieve with the bacterial sorting. Is it correct that only two bacterial subsets were sorted and sequenced – IgM coated from IgA deficient individuals and IgA coated from proficient individuals? And that noncoated in this case means not coated with IgA (in IgA proficient individuals) and not coated with IgM (in deficient individuals). Thus, IgG and/or IgM coated bacteria may be included in the non-coated fractions? The data would have had much more

interest if the authors had sorted populations based on the division used in the previous figures (i.e. sorting all the seven different groups from Figure 2 as well as non-coated bacteria).

Major point

In general, the data in Figure 1-3 are easy to understand since the data are directly coupled to the data in the manuscript. In the following figures, presented data are instead based (if I understand it right) mostly on correlations between specific properties with the proportion of bacteria that are covered in individuals. However, whether there are any causative links in these cases are unclear, and to the casual reader the way the data is presented may give the impression that experiments have been performed that actually address the question more directly.

In figure 4 the authors first determine whether different bacterial families have certain characteristics (specific localization and production of butyrate, flagellin and LPS deduced from previous publications) and compare that to whether they bound to antibodies in this study (panel A-E). They then (I think) test whether the number of bacteria in faces (total or concentration of bacteria?) in individuals correlate to the proportion of bacteria that were covered with single antibodies or combination of antibodies (proportion or concentration?). Thus, they do not test whether antibody coating with certain antibody combinations is directly correlated with having certain characteristics, but only if carrying more or less bacteria with a certain characteristic is correlated having a large fraction of bacteria covered in a certain way on an individual level. There are many possible explanations for a correlation here, but I think that most casual readers would interpret the data as that dual coating of a bacterial family is associated with it producing for example butyrate although this is not in any way proven.

In Figure 5 they then decide to study E coli further as it was the only species that produced hexa-acetylated LPS (which was strongly associated with IgG/IgM coating in IgAD but with only weakly associated with single IgA coating in healthy individuals). They then confirm that there is increased E coli presence in the IgAD (is this based on a new cohort compared to the recently published manuscript from the authors or is it in fact the same data presented again?). It is more common for IgM to coat E coli in IgAD than IgA in healthy controls (information about IgM in healthy controls is not available). Then, again, they correlate the presence E coli with the proportion of bacteria that are covered with different combinations of antibodies, and find that single IgA (in healthy individuals) or IgGIgM (in IgAD) is associated with E coli (which is not surprising since E coli is the only bacteria that produce hexa-acetylated LPS, and this feature is associated with the same coating patterns).

Overall, I think that the authors put too much emphasis on correlation analysis performed on an individual level instead of directly comparing bacterial coating patterns directly. They make statements such as

"IgG and IgM cooperate in protection of the mucosal barrier in IgA deficiency" (title)

"Since double-coating with IgM and IgG (IgMIgG) in IgA- subjects correlated strongly to bacteria carrying all the different ligands, the production of IgG, and not IgM, is presumably influenced by presence of bacteria containing immunostimulatory bacterial ligands." (results)

"Combined, this suggested that especially hexa-acylated LPS carrying bacteria may be handled by IgG in the absence of IgA" (results)

"Thus, in a complementary manner, IgG and IgM may compensate for the lack of IgA, although the increased number of E. coli in the gut of IgA- deficient subjects indicates that growth of E. coli is handled less effectively by IgM- and IgG-coating as compared to IgA-coating in IgA+ subjects, while invasiveness may be fairly controlled, as IgA- subjects experience few symptoms." (discussion)

All these statements are at this point speculative. One must assume that when antibodies are produced binding to a specific bacterial family, bacteria from that family (or a highly similar one) has triggered a response, and then flagella, LPS or other features will determine the strength or

the response rather than the proportion of bacteria having flagella in total.

I think that the only way to really demonstrate the conclusions drawn would be to sort and type bacteria for all different antibody combinations (non-coated, IgA, IgG, IgM only, dual or triple coating) from a limited number of healthy controls and IgAD individuals and see if such an experiment would confirm the inferred correlations between coating and features (i.e. that E coli are often coated with both IgM and IgG in IgAD but only IgA in healthy individuals, or whether expression of flagellin is associated with covering with multiple isotypes rather than a single one). In the absence of such sorting, the conclusions are not really supported by the data presented.

Responses to reviewer 1:

In this manuscript, Eriksen C. et al. show that IgG recognized fecal bacteria together with IgM when IgA was absent. IgG1 and IgG4 subclasses dominated bacteria-coating IgG antibodies independently of the IgA status. Of note, a significant fraction of fecal bacteria from a subset of IgA-deficient patients were coated by the IgG2 subclass. These patients also displayed increased circulating IgG and TNF -alpha, two hallmarks of inflammation possibly associated with a compromised gut barrier.

GENERAL COMMENT

Comment 1: This is a well-written manuscript that explores the impact of human IgM and IgG antibodies, including IgG subclasses, on the coating of fecal bacteria in individuals that lack IgA (i.e., in patients with IgA deficiency). Remarkably, IgA-sufficient household were used as controls, which is considered an important strength of this study. Elegantly presented data from well-performed experiments support the main conclusions. The statistical analysis of these data is considered appropriate. Weaknesses include the descriptive nature, poor mechanistic insight and limited novelty of this study. In particular, the novelty of this work is greatly attenuated by previously published reports showing binding of gut commensals by IgG and/or IgM in IgA-deficient patients and IgA-sufficient donors (refs. 6, 8, 10, 13, 17). The following additional comments are provided to enhance the impact of this manuscript.

Reply 1: We appreciate the reviewer's recognition of the efforts we put into the cohort design and analytical approaches. However, we respectfully disagree with the comment regarding the limited novelty of this study in relation to the listed references. Our intention with inclusion of the mentioned references was to emphasize the importance of a homeostatic barrier response, and it does not reflect the current insights into barrier regulation in cases of IgA deficiency. There is presently only a limited number of reports demonstrating direct IgG-coating of gut bacteria (PMID: 31771976; PMID: 30606251; PMID: 15201580; PMID: 15201580), and these studies investigate total IgG, using only relative coating frequencies, and do not compare IgA sufficient and deficient subjects. Other studies have demonstrated the presence of systemic IgG that can bind to gut bacteria (during *in vitro* settings), isolated from blood of IgA deficient individuals (PMID: 30554723), but not via analysis of *in situ* coated fecal bacteria as in this study. Differential bacterial coating with IgG subtypes is currently superficially investigated, and has only been reported in one study focusing on severe inflammation (PMID: 34404881).

Our finding that IgG cooperates with IgM in binding potential gut pathobionts in IgA deficient subjects has to the best of knowledge not been described before, and is therefore, in our opinion, both novel and of importance for understanding the function of the barrier when IgA is lacking. We therefore believe that the present findings provide important novel insights into microbe-immune dynamics in the intestine in the presence and absence of IgA, and provide significant new data describing a yet undefined role for bacterial IgG subtype coating in mucosal barrier protection. We have revised the data presented in manuscript figure 4 and 5 to bring further insight into the role played by IgM and IgG, respectively, in IgA deficiency, in the attempt to address some of the questions that were brought up during the review by reviewer 1 and 2. Please refer to the specific comments below and the revised manuscript for details.

SPECIFIC COMMENTS

Comment 2: "Gut microbiota" should be more accurately defined as "fecal microbiota". One key difference between fecal and gut commensal bacteria relates to the very low percentage of fecal microbes coated by secretory IgA (~ 7-10%) compared to gut microbes (~ 70-90%). The manuscript does not discuss this crucial difference and its potential implications.

Reply 2: We acknowledge that the use of "fecal microbiota" is a more precise definition in the current setting. This has been changed throughout the manuscript.

As a follow-up consideration, we speculate that the lower number of bacteria in the small intestine as compared to the colon (PMID: 31428414) may be the reason for the difference in relative coating between these two intestinal sites. Bunker et al. *Immunity* 2015 (PMID: 26320660) demonstrated that the % IgA coating drops significantly along the intestine in both mice and humans. In our data, we observed an inverse correlation between the load of bacteria/g feces and the % IgA coated fecal bacteria in IgA+ subjects, as stated on page 5 line 23: "An inverse correlation between the number of gut bacteria/g feces and the % of IgA coated bacteria in IgA+ subjects was observed (SCC = -0.39, $P = 0.03$, Fig 1e), indicative of a stable production of IgA during homeostatic conditions which is not increasing with increasing gut bacterial load."

The following was added to the discussion, page 13 line 4:

“In our study we found % IgA coating to inversely associate with numbers of bacteria per gram of feces, indicating that IgA may be produced in a continuous manner independently on the number of bacteria at the site. This finding is supported by previous studies on differential bacterial load along the intestinal tract, showing increasing bacterial load in the colon²⁵, implying that fecal consistency and bacterial load should be taken into consideration when comparing studies.”

Comment 3) Please, revisit the following sentence, which seems incomplete/inaccurate. “The findings regarding IgG in IgA- subjects motivated us to further investigate coating by the IgG subtypes: IgG1, IgG2, IgG3, and IgG4, which make up for the total IgG levels in the previous.”

Reply 3: Thanks for pointing this out, the sentence should read: “The findings regarding IgG in IgA- subjects motivated us to further investigate bacterial coating by the IgG subtypes: IgG1, IgG2, IgG3, and IgG4.” It has now been changed in the manuscript, page 11 line 11.

Comment 4: The increase of commensal-reactive IgG2 in a subset of IgA deficient patients is quite interesting, but should be further discussed in the context of earlier data by Benckert J. et al. (J Clin Invest, 2011). In this published study, the gut mucosa from IgA-sufficient individuals includes antigen-specific plasma cells expressing IgG1, IgG2 or IgG4 in addition to IgM, IgA1 or IgA2.

Reply 4: Thank you for pointing out this excellent study. It is very interesting that Benckert et al. (2011) find a relative distribution of IgG1+, IgG2+, IgG3+ and IgG4+ plasmablasts in the terminal ileum that is different from the distribution they see among coated fecal bacteria. The plasmablast distribution is somewhat similar to what we find for non-bacteria bound IgG1-4 fractions in feces. Moreover, Benckert and colleagues report that ca. 36% of plasmablasts expressed the *IgG2* gene and that a high proportion of the antibodies reacted largely antigen-specific to a panel of commensal and pathogenic gut bacteria, while virus-specific binding was also detected.

We have now included the following sentence in the discussion, page 11 line 20: “This finding is in line with a previous study using terminal ileum IgG+ plasmablasts, where on average 36% expressed the *IgG2* gene, that were shown to react largely antigen-specific to a panel of commensal and pathogenic gut bacteria, while virus-specific binding was also detected²⁴.”

Comment 5: Authors claim that “IgG4 is believed to be induced by the same isotype switching cytokines as IgA, namely IL-10 and TGF-beta (41),” This is inaccurate. The only universally known cytokines capable of inducing IgG4 are IL-4 and, to some extent, IL-13 (reviewed in PMID: 15229473; PMID: 18370922, PMID: 22728528; PMID: 25411432, etc.). In this regard, ref. 41 shows no data in support of the IgG4 class switch-inducing function and, more in general, the IgG4-inducing function of IL-10 and TGF-beta. Nonetheless, the involvement of these anti-inflammatory cytokines in the induction of IgG4 is plausible, particularly in the context of a “regulatory brand” of Th2 responses. However, also these responses would almost certainly require B cell-activating STAT6-inducing signals from IL-4 and/or IL-13 for the induction of IgG4.

Reply 5: Thank you very much for pointing this out. We agree that the sentence have been shortened to the extent where important details were lacking. Still, in the manuscript, we focused on human biology alone, due to the human-derived dataset, and we did not include findings from

mice in the discussion. To the best of our knowledge, the provided sources above (PMID: 15229473; PMID: 18370922, PMID: 22728528; PMID: 25411432) do not mention class-switching in humans but only in mice. Since mice, again to the best of our knowledge, do not have antibodies with the same properties as human IgG4 – especially its ability to perform Fab arm exchange – we do not consider mice to be an optimal model for investigation of these mechanisms.

Though we do agree that studies point towards IL-4 w/wo IL-13 (STAT6-induction) as important for IgG4 induction, both IL-10 and to some extent TGF-beta are important in diverging the class switching away from IgE and towards IgG4 in humans, as demonstrated in several studies (PMID: 9531318; PMID:18925882; PMID:28782655; PMID:18924213; PMID:16998509).

We have now changed the phrasing in the discussion to, page 14 line 22: “In humans, IgG4 responses are believed to be enhanced by the same isotype switching cytokines as IgA, namely IL-10 and TGF- β ³⁸ in the presence of IL-4. IL-10 and TGF- β are regarded as anti-inflammatory cytokines produced during homeostatic conditions in the gut.”

Comment 6: The homeostatic function of IgG4 in the gut mucosa may be independent of non-inflammatory environmental signals (i.e., IL-10 and TGF-beta) required for its induction and rather be linked to the poorly pro-inflammatory nature of the heavy chain Cgamma4 region of IgG4. Indeed, Cgamma4 does not have any prominent pro-inflammatory effector function, being unable to engage activating Fcgamma receptors or initiate the complement cascade. These crucial biological aspects of IgG4 are not discussed.

Reply 6: We absolutely agree that this an important, if not the most important, biological function of IgG4. We did not include this in the first place as the study was not focused on IgG4, but on differences between coating patterns in IgA+ and IgA- subjects, where IgG4 coating showed to be similar (and then there is a word count restriction to take into consideration). However, we have now added a part to the discussion to clarify this important point (page 14 line 25), “IgG4 might act as a mediator for tolerance via its low affinity binding to activating compared to inhibitory Fc γ receptors and its inability to initiate the complement cascade³⁹. In addition, IgG4 is able to perform Fab arm exchange in vivo⁴⁰, which combine specificities of two IgG4 molecules effectively preventing crosslinking of antigens. As it was not the main focus of the current study to elucidate mechanisms of IgG1 and IgG4, we can only speculate that one or both of them may be involved in processes similar to that of IgA, hence assisting in maintaining an inner mucus layer relatively free of microorganisms without giving rise to excessive immune reactions towards the commensal members of the microbiota⁴¹.”

Comment 7: No direct evidence is provided to show the protective activity of IgM and IgG on the mucosal barrier in IgA deficiency. In particular, no direct evidence is provided to show the perturbation of the intestinal barrier in IgA-deficient patients with increased IgG2-coated bacteria as well as circulating IgG and TNF -alpha.

Reply 7: We agree that it would be fantastic to provide evidence for a protective role of double-coating of bacteria with IgM and IgG within the mucosal barrier in IgA deficiency. Unfortunately, we can only speculate on the role of double-coating of bacteria with IgM and IgG, as the implications, as we see it, can only be demonstrated in double IgG- and IgA-deficient individuals, which to the best of our knowledge have not been described. We have added new data as revised Fig. 4A and 4B (figures inserted below) that demonstrate a high prevalence of bacteria that are

double-coated with IgM and IgG across most bacterial families in IgA deficient subjects, which is not seen to the same extent for IgA and IgG in IgA sufficient subjects, where single IgA coating is more dominant. In our opinion, this signifies that IgG's are co-assisting IgM in coating of most gut bacterial families when IgA is lacking, while bacterial co-coating with IgG and IgA is less predominant in IgA sufficient subjects. Moreover, the fact that IgA deficient subjects generally are asymptomatic might indirectly support the notion of a protective activity of IgM and IgG on the mucosal barrier in IgA deficiency.

Figure 4. IgG assists IgM in coating of bacteria in the absence of IgA.

(a) Heatmap showing % prevalence of coated bacterial families within FACS sorted single-IgA and double-IgAIgG coated bacteria from IgA+ subjects, and single-IgM and double-IgMIgG coated bacteria from IgA- subjects. Bacteria were identified upon 16S rRNA gene amplicon sequencing identifying uniquely coated bacteria based on amplicon sequence variants (ASVs). Grey boxes represent families with no identified Ig-coating. (b) Overall prevalences of Ig-coated bacteria for each Ig combination from (a). Wilcoxon test was used to compare the coating prevalence amongst all Ig combinations, with *, $P < 0.05$, **, $P < 0.01$, ***, $P < 0.001$. For boxplots, the center line indicates the median and the box limits indicate the quartiles. Whiskers extend to the data points within 1.58x the interquartile range, and outliers are shown as individual dots.

We have added a section to the results section describing the data in Fig. 4A and 4B, to the discussion section, and the methods section:

Page 8 line 19 (results part),

“section header: IgG assists IgM in coating of most bacterial families in IgA deficiency.

We next aimed to determine the extent of coating of bacterial families with IgG together with IgM in IgA- subjects, and with IgA in IgA+ subjects. This was done by FACS-sorting single IgA as well as double-IgAIgG coated bacteria from IgA+ subjects, and single IgM as well as double-IgMIgG coated bacteria from IgA- subjects. The prevalence of bacterial coating with single IgM and double-IgMIgG in IgA- subjects was found to be similar, while the prevalence of bacterial coating with single IgA was significantly higher than that of double-IgAIgG (Fig 4a, b). This implies that IgG coating is more frequently assisting IgM in coating of bacteria in IgA- subjects than is the case for IgG and IgA in IgA+ subjects.”

Page 14 line 8 (discussion part),

“Thus, in a complementary manner, IgG and IgM may compensate for the lack of IgA, although the increased load of *E. coli* in feces of IgA-deficient subjects indicates that growth of *E. coli* is handled less effectively by IgM and IgG coating as compared to IgA coating in IgA+ subjects, while invasiveness may be fairly controlled, as IgA- subjects experience few symptoms⁴.”

Page 15 line 12 (discussion),

“In the presence of IgA, single-coating with IgA was mostly prominent, while, single- and double-coating of bacteria with IgM and IgG dominated in the absence of IgA. This switch to bacterial double-coating in IgA deficiency could be one of the reasons for the partial bacterial-specific barrier defense experienced in IgA-deficient humans, and explain why they experience relatively mild disease symptoms over a life time⁴.”

Page 17 line 5 (methods),

“For a subgroup of subjects we also sorted specifically single-IgA, -IgM and -IgG in addition to double-IgAIgG and double-IgMIgG coated bacteria.”

We agree that the part on bacterial single IgG2-coating and its relation to increased circulating TNF- α and IgG (pro-inflammation) would benefit from additional support. Despite the general asymptomatic living of IgA deficient subjects, it has for many years been acknowledge that some IgA deficient subjects display increased influx of macromolecules, representing enhanced gut permeability (PMID: 2387510). Enhanced gut permeability is related to enhanced levels of circulating TNF-alpha and IgG (PMID: 11156641; PMID: 26944199). Such interrelationship has also been extensively reported for IgA sufficient subjects, especially in individuals with intestinal inflammation (PMID: 11156641). In this study, we observed a novel interrelationship adding to these previous findings, namely an increased correlation between single-IgG2 coating vs. fecal *E. coli* load, and enhanced levels of the circulating pro-inflammatory factor TNF-alpha in individuals with single-IgG2 bacterial coating, hence pointing to a role of bacterial coating with single IgG2 to associate with increased gut permeability.

We have not been able to recover intestinal biopsies from IgA deficient subjects with and without IgG2 coating, and therefore, we cannot provide direct evidence for increased gut permeability in IgA deficient subjects. We have therefore toned down the discussion on implications related to single IgG2 bacterial coating, circulating IgG and gut permeability. This has led to removal of original Fig. 5J and 5K, showing relations to serum IgG, from revised Fig. 6, and removal of text related to IgG in the manuscript text.

Instead, we have included new data comparing the prevalence of individuals with single IgG2 coating in IgA+ and IgA- subjects to data from subjects with remissive and active Crohn's disease-based gut inflammation (new Fig. 6I, see next page). Data show >1.5-fold increase in the number of subjects with single-IgG2 coated bacteria in IgA- vs. IgA+ subjects, while it is >2-fold higher in subjects with active gut inflammation (IgA+: 5 out of 31 = 16.1%, IgA-: 8 out of 31 = 25.8%, remissive Crohn's disease: 14 out of 33 = 42.4%, active Crohn's disease: 16 out of 27 = 59.3%).

Combined with the original Fig. 5 data, now revised Fig. 6, showing increased circulating levels of the pro-inflammatory cytokine TNF- α in individuals with single-IgG2 bacterial coating, the overall data in revised Fig. 6 demonstrate that single-IgG2 bacterial coating is a marker of inflammation that is linked to *E. coli* fecal load, and occurs with increased prevalence in IgA- vs. IgA+ subjects, but without reaching the levels found in Crohn's disease patients having either remitting or active

gut inflammation. Based on the higher prevalence seen in Crohn's disease patients, we speculate that appearance of single-IgG2 bacterial coating may be a phenomenon related to gut inflammation, increased permeability, resulting in enhanced systemic TNF- α . We also sorted out IgG2 coated bacteria and confirmed that *E. coli* is indeed IgG2 coated, but since it was only possible to obtain enough material for IgG2 sorting in Crohn's disease patients with severe disease, the sorting data has not been added to the current manuscript, where the focus remains to be on bacterial coating in IgA- vs. IgA+ subjects.

Figure 6. IgG assists IgM in coating of *E. coli* in the absence of IgA.

(i) Prevalence of subjects with single-IgG2 coating in Crohn's disease (CD) patients during active disease or remission compared to IgA- and IgA+ subjects. Differences in single-IgG2 prevalence between between CD and the IgA cohort was tested using Fisher's exact test.

Due to the added changes to revised Fig. 6, we have revised the text in the abstract, results and discussion part accordingly.

Page 2 line 2 (abstract),

"Immunoglobulin A (IgA) is acknowledged to play a role in the defense of the mucosal barrier by coating microorganisms. Still, IgA-deficient humans exhibit few infection-related complications raising the question if the more specific IgG may help IgM in compensating for the lack of IgA. We here used a cohort of 31 IgA-deficient humans each paired with IgA-sufficient household members to investigate multi-Ig bacterial coating. In IgA-deficient humans, IgM alone, and together with IgG, recapitulate coating of most bacterial families, despite an overall 3.6-fold lower Ig-coating. Bacterial IgG coating was dominated by IgG1 and IgG4. Single-IgG2 bacterial coating was sparse, 1.6-fold more prevalent in IgA deficiency, and linked to enhanced *Escherichia coli* load and TNF- α , although 2-fold less prevalent than in inflammatory bowel disease. Altogether we demonstrate that IgG assists IgM in coating of most bacterial families in the absence of IgA, and identify single-IgG2 bacterial coating as an inflammatory marker."

Page 12 line 6 (results part),

"Abundances of IgG1, IgG2 and IgG4 single-coated gut bacteria correlated with a SCC of 0.33, 0.78, and 0.32, respectively to the total fecal *E. coli* load across all subjects (Fig 6g, Supplementary

Table 7). For IgG1 and IgG4 this was comparatively similar to the SCC of 0.46 seen for single-IgA coating vs. *E. coli* load (Supplementary Table 7). This suggests that bacterial coating with IgG1 and IgG4 might play a role as barrier-maintaining factors in healthy subjects. In subjects displaying single-IgG2 bacterial coating (21% of subjects), we found higher systemic levels of the pro-inflammatory cytokine TNF- α than in subjects with no single-IgG2 bacterial coating (Fig 6h, $P < 0.01$). Although single-IgG2 bacterial coating was relatively sparse, we found it to be 1.6-fold more prevalent in IgA- vs. IgA+ subjects, whilst prevalence increases by an additional 2-fold in Crohn's disease patients (Fig 6i). This points to single-IgG2 bacterial coating to be enhanced during inflammatory processes.”

Page 14 line 12 (discussion part),

“However, we find a 1.6-fold increased prevalence of IgA- vs. IgA+ subjects with single-IgG2 bacterial coating. Factors promoting isotype switching to IgG2 are largely unexplored³³, but it is reported that IgG may be induced by the type 1 immune cytokines IFN- γ , IL-12 and IL-18^{34,35}, by specific pathogenic microbes^{34,35}, and in response to *E. coli*-derived LPS stimulation in vitro³⁷, thereby supporting a link between the presence of hexa-acylated LPS-producing bacteria, such as *E. coli*, and IgG2 production. Enhanced systemic TNF- α levels in subjects with single-IgG2 coated bacteria, and increased prevalence in Crohn's disease patients with active disease, further substantiate that single-IgG2 coating is linked to local and systemic inflammation, and may be a marker for on-going inflammatory processes.”

Page 15 line 9 (discussion part),

“IgA-deficient subjects displayed 1.6-fold higher prevalence of single-IgG2 bacterial coating that linked to fecal *E. coli* load and concomitantly increased TNF- α levels in blood. Single-IgG2 coating was even more prevalent in Crohn's disease patients, hence suggesting a role for single-IgG2 coating during inflammatory conditions.”

Page 16 line 1 (methods part),

“Samples for comparison to single-IgG2 coating prevalence in Crohn's disease patients were derived from the IBD South Limburg (IBDSL) cohort, which is a population-based inception cohort from the South Limburg area of the Netherlands, collected as described⁴², approved by the local Medical Ethics Committee, and registered in <http://www.clinicaltrials.gov> (NCT02130349).”

Response to Reviewer 2:

Comment 1: In their manuscript “IgG and IgM cooperate in protection of the mucosal barrier in IgA deficiency”, Eriksen and coworkers study antibody covering of the gut microbiota in IgA proficient and deficient (IgAD) individuals from the same households. They find that IgM substitute for IgA in IgAD, that IgG is more often covering bacteria than previous studies have suggested, although they do not see differences depending on IgA status. Dual or triple coating by several Ig classes is not uncommon. They compare the bacterial composition of IgM coated species (from IgA deficient individuals) with IgA coated species (from IgA proficient individuals). Based on previously published datasets, they determine the likely localization of antibody coated and non-coated bacteria, and found that presence of flagellin and potential to produce acetylated LPS was associated with IgA coating, while this was less pronounced for butyrate production. E coli numbers increased in IgA deficient individuals, in which E. coli was also more often covered with IgM in deficient individuals than with IgA in proficient ones. IgG1 and IgG4 were the most prevalent classes coating gut bacteria, while IgG2 was present in faces but did not coat bacteria.

In general, this is a relatively well written report of experiments performed on IgA deficient individuals, the largest immunodeficient group. As these in general have a rather mild phenotype despite lacking the antibody class that is produced in largest quantities, it is important to study to which extent compensatory mechanism (such as secretion of other antibody classes) may explain this. Relatively few studies have been presented that study this in depth, and for this reason the study is of interest.

Strengths of the study

The authors study a relatively large cohort of IgA proficient and deficient individuals from the same households to diminish variability due to selection

The authors demonstrate that IgM coating appear to take over the coating role of IgA in the absence of it.

The authors find that IgG coating from IgG1 and IgG4 antibodies is more prevalent than previously thought, but that IgG appear to play a minor role in substituting for IgA.

Weaknesses

Other studies have been published that have studied the composition of IgA bound vs IgM bound in IgA deficient individuals, so the study is not unique in this manner.

Data from individuals from the same households are not used for paired analysis, which may have revealed additional information. In the absence of pairing, it is unclear whether the use of household samples will in fact decrease variability for each group.

Reply 1: We appreciate the reviewer’s recognition of strengths, as well as weaknesses, of our study. We have a few clarifications to add on parts that may not have been sufficiently clear, both in terms of weaknesses and strengths:

In relation to the paired analysis: All comparative analyses between the IgA deficient individuals and their partners were performed using paired analysis. By this comment, we realized that this was not sufficiently clear from the method section/figure legends of the manuscript. We have now added to Page 19 line 9: “Paired Wilcoxon rank-sum test statistics were used when comparing IgA deficient subjects with partners” to the statistical analysis section in methods, and this has moreover been specified in appropriate figure legends.

In relation to the comments regarding novelty, we realized that we did not sufficiently emphasize the novelty of the findings in our conclusions, especially regarding the findings of the role of IgG in double-coating of bacteria. We observed similar numbers of double-coated bacteria with IgAIgG in IgA+, and IgMIgG in IgA- subjects (Fig. 2B), but coating with IgMIgG was found to be more prevalent across most bacterial families in IgA- subjects than seen for IgAIgG in IgA+ subjects (data added as new Fig. 4A and 4B to displace earlier correlations in original Fig. 4F).

Figure 2. Frequency of single-, double- and triple-Ig coated fecal bacteria in the presence and absence of IgA. (b) Average distribution of coated gut bacteria (in gram per feces) in IgA+ and IgA- subjects, n = 31 in each group. Paired Wilcoxon test was used to compare IgA- and IgA+ pairs, with *, $P < 0.05$, **, $P < 0.01$, ***, $P < 0.001$, ****, $P < 0.0001$. For boxplots, the center line indicates the median and the box limits indicate the quartiles. Whiskers extend to the data points within 1.58x the interquartile range, and outliers are shown as individual dots.

Figure 4. IgG assists IgM in coating of bacteria in the absence of IgA.

(a) Heatmap showing % prevalence of coated bacterial families within FACS sorted single-IgA and double-IgAIgG coated bacteria from IgA+ subjects, and single-IgM and double-IgMIgG coated bacteria from IgA- subjects. Bacteria were identified upon 16S rRNA gene amplicon sequencing identifying uniquely coated bacteria based on amplicon sequence variants (ASVs). Grey boxes represent families with no identified Ig-coating. (b) Overall prevalences of Ig-coated bacteria for each Ig combination from (a). Wilcoxon test was used to compare the coating prevalence amongst all Ig combinations, with *, $P < 0.05$, **, $P < 0.01$, ***, $P < 0.001$. For boxplots, the center line indicates the median and the box limits indicate the quartiles. Whiskers extend to the data points within 1.58x the interquartile range, and outliers are shown as individual dots.

This is pointing to IgG bacterial coating as an additional layer of support in maintaining the mucosal barrier response in both IgA deficient and sufficient subjects, while we speculate that double-coating with IgM and IgG may play a greater role in IgA deficiency due to the double-coating identified for most bacterial families and due to its higher prevalence across subjects. Based on this, we have updated the manuscript to emphasize the added novel findings to the manuscript:

Abstract: Page 2 line 6, "In IgA-deficient humans, IgM alone, and together with IgG, recapitulate coating of most bacterial families, despite an overall 3.6-fold lower Ig-coating."

Line 10, "Altogether we demonstrate that IgG assists IgM in coating of most bacterial families in the absence of IgA,"

Page 8 line 19 (results part),

"section header: IgG assists IgM in coating of most bacterial families in IgA deficiency.

We next aimed to determine the extent of coating of bacterial families with IgG together with IgM in IgA- subjects, and with IgA in IgA+ subjects. This was done by FACS-sorting single IgA as well as double-IgAIgG coated bacteria from IgA+ subjects, and single IgM as well as double-IgMIgG coated bacteria from IgA- subjects. The prevalence of bacterial coating with single IgM and double-IgMIgG in IgA- subjects was found to be similar, while the prevalence of bacterial coating with single IgA was significantly higher than that of double-IgAIgG (Fig 4a, b). This implies that IgG coating is more frequently assisting IgM in coating of bacteria in IgA- subjects than is the case for IgG and IgA in IgA+ subjects."

Page 14 line 8 (discussion part),

"Thus, in a complementary manner, IgG and IgM may compensate for the lack of IgA, although the increased load of *E. coli* in feces of IgA-deficient subjects indicates that growth of *E. coli* is handled

less effectively by IgM and IgG coating as compared to IgA coating in IgA+ subjects, while invasiveness may be fairly controlled, as IgA- subjects experience few symptoms⁴.”

Page 15 line 12 (discussion),

“In the presence of IgA, single-coating with IgA was mostly prominent, while, single- and double-coating of bacteria with IgM and IgG dominated in the absence of IgA. This switch to bacterial double-coating in IgA deficiency could be one of the reasons for the partial bacterial-specific barrier defense experienced in IgA-deficient humans, and explain why they experience relatively mild disease symptoms over a life time⁴.”

Page 17 line 5 (methods),

“For a subgroup of subjects we also sorted specifically single-IgA, -IgM and -IgG in addition to double-IgAIgG and double-IgMIgG coated bacteria.”

Comment 2: It appears to be a follow up study on IgA deficient individuals from the same group (Moll et al. Gastroenterology 7: 2423). The overlap (if any) between the cohorts in previous and this study is not clear. Some of the findings (i.e. increased levels of *E. coli* in IgA deficient individuals) were reported in the previous study.

Reply 2: The herein used IgA deficient/sufficient household pairs are a subsample of the cohort used in Moll et al., as explained in the cohort design section in Methods. The current study is based on samples from 31 out of the 50 household pairs from which sufficient material was available to conduct Ig-coating analyses. The metagenomic sequencing data used in this study to conduct the *in silico* functional analyses in revised Fig. 5, and for revised Fig. 6A *E. coli* information is publicly available (see data availability), and released with publication of Moll et al. The *in silico* functional analyses of butyrate and flagellin production capability (based on gene presence within metagenomic species) are completely new, and the coupling of these to the coated bacterial families in revised Fig. 5 is notoriously also new. It is correct that we reported the *E. coli* prevalence in Moll et al. for the 50 household pairs. Revised Fig. 6A data are based on the 31 pairs, and we believe it is necessary to include the data for the current study population, before we investigate relations between *E. coli* and coating patterns. The latter was not part of Moll et al.

We have now further clarified this part within the study design section in Methods and in the results section,

Page 15 line 23: “The current study is based on samples from 31 out of 50 household pairs for which sufficient material was available to conduct Ig-coating analyses.”

Page 10 line 24: “We found *E. coli* numbers to be 7.4-fold higher in the 31 household pairs of IgA- as compared to IgA+ subjects in the present study (Fig 6a, $P = 0.012$). This finding is reported earlier in a larger subgroup of the same cohort¹⁵.”

Comment 3: It is unclear what the authors try to achieve with the bacterial sorting. Is it correct that only two bacterial subsets were sorted and sequenced – IgM coated from IgA deficient individuals and IgA coated from proficient individuals? And that noncoated in this case means

not coated with IgA (in IgA proficient individuals) and not coated with IgM (in deficient individuals).

Thus, IgG and/or IgM coated bacteria may be included in the non-coated fractions? The data would have had much more interest if the authors had sorted populations based on the division used in the previous figures (i.e. sorting all the seven different groups from Figure 2 as well as non-coated bacteria).

Reply 3: Thanks for pointing out this misunderstanding. The term non-coated bacteria was used to annotate bacteria that were found to be consistently non-coated with any of the antibody isotypes in both IgA deficient and sufficient subjects, so they are not coated with IgA, IgM or IgG (i.e. the bacteria annotated as non-coated are triple-negative for IgA, IgM and IgG). These bacteria were sorted out from each subject. We agree that this may not have been sufficiently clear in the previous version, and have now specified this in the methods and result sections:

Page 7 line 6: “We also sorted out bacteria that were triple-negative for IgA, IgM and IgG (non-coated) from each of these subjects to specifically define coated and non-coated bacteria per individual.”

Page 17 line 7: “From all subjects, we also sorted out non-coated bacteria (triple-negative for IgA, IgM and IgG).”

It is correct that the displayed sequencing data are generated from sorting of the bacterial subsets coated with any IgA combination (which contain single-IgA, double-IgAIgM and double-IgAIgG coated bacteria) in IgA+ subjects and any IgM combination (including single-IgM and double-IgMIgG coated bacteria) in IgA- subjects. Our aim here was not to identify individual bacteria coated with these five different sub-Ig-combinations. Our first aim was to characterize the multi-coating profiles within IgA- and IgA+ subjects. We next aimed to compare the taxa coated with any IgM (single or multi) in IgA-subjects vs. those coated with any IgA (single or multi) in IgA+ subjects for each household pair. The reason for this was that the general focus of the study was on the collective protection of the mucosal barrier by Ig-coating of bacteria in IgA- vs. IgA+ subjects, and not on the coating patterns of specific taxa, although the latter is also very interesting. We are sorry if the general focus of our study was not sufficiently clear in the original version in regards to the strength of the study design, where we included **paired analysis** of the **load of single- and multi-coated** and **strictly non-coated bacteria** to identify the differences in coating of bacterial taxa within IgA- vs. IgA+ household pairs with high confidence. We have now changed the phrasing around the sorting part to:

Page 7 line 2: “We next used a FACS-based sorting approach⁹ to sort out all IgA coated bacteria in IgA+ subjects and all IgM coated bacteria in IgA- subjects to identify the taxonomy and coating frequency of single- and multi-coated bacteria with any IgA or IgM, respectively.”

Due to your comments regarding single-Ig coated vs. double-Ig coated bacteria, we have performed and added additional experimental data to the manuscript, as new Fig. 4. In those we sorted out single-IgA and double-IgAIgG coated bacteria from IgA+ subjects and single IgM and double-IgMIgG coated bacteria in IgA- subjects for comparison of the prevalence of bacterial coating with each of these Ig combinations in IgA- vs. IgA+ subjects. The decision for sorting of these specific Ig combinations was based on data in original Fig. 2, which shows that that single-

IgA and double-IgA/IgG and single IgM and double -IgM/IgG coating were the major discriminating coating types between IgA+ and IgA- subjects. Please refer to reply to comment 5 below for additional information.

Major point

Comment 4: In general, the data in Figure 1-3 are easy to understand since the data are directly coupled to the data in the manuscript. In the following figures, presented data are instead based (if I understand it right) mostly on correlations between specific properties with the proportion of bacteria that are covered in individuals. However, whether there are any causative links in these cases are unclear, and to the casual reader the way the data is presented may give the impression that experiments have been performed that actually address the question more directly.

Reply 4: It is correct that the data presented in the original Fig. 4C-F were partly derived from *in silico*-based prediction of functional capabilities of the individual bacterial taxa that were identified by FACS sorting of coated bacteria (Fig. 3 data). We believe such *in silico* analyses are necessary in order to move the field forward until all single strains can be cultivated from human feces, and thereby functionally investigated. We have tried to be cautious in how it is phrased throughout the manuscript, using terms such as ‘immunostimulatory potential’ and ‘*in silico*’. Upon considerable revision of the data in mainly Fig. 3 and revised Fig. 5 according to reviewer comments, we have decided to keep the *in silico* analysis (new Fig. 5C), but have revised it to contain all the families that appeared upon re-analyzing Fig. 3 data. We have moreover included a new Fig. 5D comparing the ratio of Ig-coated *in silico* predicted producers of flagellin, hexa-acylated LPS, penta-acylated LPS or butyrate to Ig-coated non-producers in IgA- and IgA+ paired subjects. This analysis revealed that IgA- subjects have a significantly higher proportion of Ig-coating of bacteria predicted to produce flagellin and hexa-acylated LPS than IgA+ subjects. It also revealed that predicted producers are not preferentially more Ig-coated than non-producers, as the ratio is below 1 for all predicted products.

We have decided to delete the previous correlation analyses in original Fig. 4F, as we agree with the comment that it may be misinterpreted by the casual reader. In the revised manuscript, we have now included additional data on coating to directly answer some of the comments below. Please refer to our answer for comment 5 below.

Comment 5: In figure 4 the authors first determine whether different bacterial families have certain characteristics (specific localization and production of butyrate, flagellin and LPS deduced from previous publications) and compare that to whether they bound to antibodies in this study (panel A-E). They then (I think) test whether the number of bacteria in feces (total or concentration of bacteria?) in individuals correlate to the proportion of bacteria that were covered with single antibodies or combination of antibodies (proportion or concentration?).

Thus, they do not test whether antibody coating with certain antibody combinations is directly correlated with having certain characteristics, but only if carrying more or less bacteria with a certain characteristic is correlated having a large fraction of bacteria covered in a certain way on an individual level. There are many possible explanations for a correlation here, but I think that most casual readers would interpret the data as that dual coating of a bacterial family is associated with it producing for example butyrate although this is not in any way proven.

Reply 5: First of all, for clarification, the deduction of the ability to produce the immunoregulatory ligands flagellin and butyrate in bacterial species was performed as part of the current study. The information for hexa- and penta-acylated LPS production in bacterial species was published in Moll et al. *Gastroenterology* 7: 2423 (2021).

We agree with the reviewer that the correlation analysis in original Fig. 4F may be confusing to most readers, and have now conducted additional experiments and analyses that have replaced original Fig. 4F, as suggested by the reviewer. We thank you very much for this comment, as the current information is what was necessary to understand the coating dynamics in IgA- vs. IgA+ subjects.

Included changes:

In revised Fig. 4C, we have added the actual coating frequency of bacteria with a given *in silico* predicted potential. The data are now shown in a combined dual-triangle plot, where the upper triangle information is similar to the information provided in original Fig. 4C-E, whereas the actual coating frequency of bacteria with the predicted immunostimulatory ligand is now provided in the lower triangle. We have moreover compiled the information across all bacteria in the added and revised Fig. 5D to compare the ratio of Ig-coated *in silico* predicted producers of flagellin, hexa-acylated LPS, penta-acylated LPS or butyrate to Ig-coated non-producers within IgA- and IgA+ paired subjects. As mentioned earlier, this analysis revealed that IgA- subjects have a significantly higher proportion of Ig-coating of bacteria predicted to produce flagellin and hexa-acylated LPS than IgA+ subjects. This corresponds to higher relative coating in IgA-subjects of bacteria within flagellin-producing bacterial families of *Clostridiaceae*, *Lachnospiraceae*, *Peptostreptococcaceae*, *Ruminococcaceae*, *Desulfovibrionaceae*, and *Enterobacteriaceae*, and for hexa-acylated LPS-producing bacteria within *Enterobacteriaceae*. Moreover, in analyses conducted for the revision and added as new Fig. 4A and 4B, we find that single-IgM and double-IgMIgG coating of all these bacterial families is almost alike amongst IgA- subjects, while this is not the case for single-IgA and double-IgAIgG in IgA+ subjects. We therefore conclude that IgM bacterial coating in IgA deficiency is supported by double-coating with IgM and IgG.

Please refer to fig 4 in reply 1.

Please refer to reply 1 and 3 for the added manuscript text parts.

Comment 6: In Figure 5 they then decide to study *E coli* further as it was the only species that produced hexa-acetylated LPS (which was strongly associated with IgG/IgM coating in IgAD but with only weakly associated with single IgA coating in healthy individuals). They then confirm that there is increased *E coli* presence in the IgAD (is this based on a new cohort compared to the recently published manuscript from the authors or is it in fact the same data presented again?).

Reply 6: As explained in reply 2, we decided to include *E. coli* abundance for the herein studied subgroup (N=31 here, N= 50 in Moll et al.) in order to provide the study relevant data in the manuscript. This has now been better explained in the Results and Methods sections, as described in reply 2.

Comment 7: It is more common for IgM to coat *E. coli* in IgAD than IgA in healthy controls (information about IgM in healthy controls is not available). Then, again, they correlate the presence *E. coli* with the proportion of bacteria that are covered with different combinations of antibodies, and find that single IgA (in healthy individuals) or IgGIgM (in IgAD) is associated with *E. coli* (which is not surprising since *E. coli* is the only bacteria that produce hexa-acetylated LPS, and this feature is associated with the same coating patterns).

Reply 7: We agree that the data in original Fig. 5C for *E. coli* are not very surprising taken the data in original Fig. 4F into consideration. In the revision, we have decided to remove the original Fig. 4F as per suggestion by the reviewer, and have kept the data in Fig. 5C (now revised Fig. 6C), as we find it brings important value by demonstrating the difference between the fecal *E. coli* concentration and its correlation to the number of double-IgGIgM coated bacteria and not to single-IgM coated bacteria in IgA- subjects, while to single-IgA, and not to double-IgAIgG in IgA+ subjects. Since, in our data the *Enterobacteriaceae* family also encompass ASVs with annotation to *Enterobacteriaceae_f*, as well as *Klebsiella* and *Enterobacter*, besides *E. coli*, we find it relevant to include the correlations for *E. coli*, although it is the most prevalent gut pathobiont in the *Enterobacteriaceae* family.

Comment 8: Overall, I think that the authors put too much emphasis on correlation analysis performed on an individual level instead of directly comparing bacterial coating patterns directly. They make statements such as

"IgG and IgM cooperate in protection of the mucosal barrier in IgA deficiency" (title)

"Since double-coating with IgM and IgG (IgMIgG) in IgA- subjects correlated strongly to bacteria carrying all the different ligands, the production of IgG, and not IgM, is presumably influenced by presence of bacteria containing immunostimulatory bacterial ligands." (results)

"Combined, this suggested that especially hexa-acetylated LPS carrying bacteria may be handled by IgG in the absence of IgA" (results)

"Thus, in a complementary manner, IgG and IgM may compensate for the lack of IgA, although the increased number of *E. coli* in the gut of IgA- deficient subjects indicates that growth of *E. coli* is handled less effectively by IgM- and IgG-coating as compared to IgA-coating in IgA+ subjects, while invasiveness may be fairly controlled, as IgA- subjects experience few symptoms." (discussion)

All these statements are at this point speculative. One must assume that when antibodies are produced binding to a specific bacterial family, bacteria from that family (or a highly similar one) has triggered a response, and then flagella, LPS or other features will determine the strength or the response rather than the proportion of bacteria having flagella in total.

Reply 8: We agree with this comment, and acknowledge that additional analyses that directly examine bacterial coating with single-IgM vs. double-IgMIgG in IgA- subjects were needed. We have now performed these experiments, and included the results as new Fig. 4A and 4B, as explained in detail in reply 1 and 3.

Fig. 4A and 4B data revealed that in IgA⁻ subjects most of the bacterial families that are coated with IgM hold concomitant coating with IgG, while in IgA⁺ subjects, the bacterial families are more often only coated with IgA. The prevalence of coating is shown per bacterial family (Fig. 4A) and overall (Fig. 4B). Based on these data, we have decided to propose that we stick to the message in the original title (IgG and IgM cooperate in...), as the additional data support this conclusion. However, we agree to revise it to more specifically highlight that it was coating of intestinal bacteria we have addressed and not protection of the mucosal barrier.

The title has been changed to “IgG and IgM cooperate in coating of intestinal bacteria in IgA deficiency”. We hope you agree with this modification.

Comment 9: I think that the only way to really demonstrate the conclusions drawn would be to sort and type bacteria for all different antibody combinations (non-coated, IgA, IgG, IgM only, dual or triple coating) from a limited number of healthy controls and IgAD individuals and see if such an experiment would confirm the inferred correlations between coating and features (i.e. that E coli are often coated with both IgM and IgG in IgAD but only IgA in healthy individuals, or whether expression of flagellin is associated with covering with multiple isotypes rather than a single one). In the absence of such sorting, the conclusions are not really supported by the data presented.

Reply 9: Thanks for this suggestion. We would first like to emphasize that the data presented in the original manuscript were already based on comparisons to consistently non-coated bacteria across all sorted and sequenced samples. We have now strengthened this in the data representation by stating strictly non-coated in Fig. 3, and emphasized it in the respective text parts, as earlier clarified in reply 3.

Since individuals overall differ in their bacterial composition, and also in their overall coating patterns, it is difficult to make firm general conclusions on coating of individual ASVs and species depending on their *in silico* immunostimulatory potential. First of all, a given bacterium needs to be present in a given subject to get coated. Since the presence, abundance, and % coating of bacteria vary across individuals, we have decided to provide the average coating pattern in Fig. 3D across the sorted and sequenced samples (16 IgA⁻ vs. 16 IgA⁺ household pairs). For the *in silico*-inference analysis, we indeed did the analysis as suggested above, although our focus was alone on any IgA and any IgM coating, in IgA⁺ and IgA⁻ subjects, respectively. However, we agree that the way this was visualized in the original Fig. 4C-D was not really conveying this message. We have now visualized data differently, presenting it in a dual-triangle format in revised Fig. 5D, where the upper triangle provides the same information as in the previous Fig. 4C-D (% of bacteria with the inferred potential per bacterial family), while the lower triangle provides information on the identified % frequency of coating of bacteria with a given potential. As explained earlier, in order to provide information related to the general coating frequencies of bacteria with a given potential vs. those that do not hold this potential, we have now included new Fig. 5D showing the ratio of the average coating frequency amongst bacteria with the inferred potential vs. the average coating frequency of bacteria without this potential.

It was not our intention to unfold IgG-coating patterns for bacteria with certain potentials in this study, although it is indeed interesting. The reason is due to the existence of the four IgG isotypes that may well be differentially enhanced based on stimulating ligands. So, in order to fully address

such aspect, one would need to provide data on each of the isotypes, which would require huge amounts of fecal material per subject, due to the varying coating-% per IgG isotype. For this specific cohort of household-paired IgA+ and IgA- subjects, we do not have this amount of available sample, but it may be possible in future studies using other cohorts.

On behalf of the authors,
Carsten Eriksen and Susanne Brix

REVIEWER COMMENTS

Reviewer #1 (Remarks to the Author):

GENERAL COMMENT

Both clarity and overall insight of this manuscript have improved following revision. Novel aspects are better emphasized and some interpretations clarified. Notwithstanding these improvements and the relative novelty of some findings (i.e., IgG subclass binding profiles and composition of gut microbes from IgA deficient patients; identification of bacteria from these patients dually coated by IgG and IgM), this report transmits a "dejà vu" feeling that stems with some overlap with the existing literature (e.g., ref. 13). The lack of antibody binding data from bacteria collected from the intestine further attenuates my enthusiasm for an otherwise interesting and nicely executed study. Some key literature is also missing. That being said, I praise authors for their efforts. Again, the manuscript has certainly improved.

SPECIFIC COMMENTS

1) Some of the findings of this work are incremental at best. This is the case of data showing minimal IgG binding to fecal bacteria in healthy controls. In their rebuttal letter, authors state: "There is presently only a limited number of reports demonstrating direct IgG-coating of gut bacteria (PMID: 31771976; PMID: 30606251; PMID: 15201580; PMID: 15201580) ..." Aside from inadvertently duplicating PMID: 15201580, authors omit PMID: 30876876, which compares IgG+ fecal bacteria from household healthy controls with IgG+ fecal bacteria from ulcerative colitis patients. PMID: 30876876 should be quoted and findings from its Fig. 1B briefly discussed, also in relationship to PMID: 30554723.

2) In PMID: 30554723 (ref. 13), there is a virtual lack of IgG+ bacteria in the stool from healthy individuals, which contrasts with the much higher frequency of these bacteria in the present study. Can authors elaborate on possible reasons behind this striking discrepancy?

3) Line 22 of the revised Discussion is more accurate than its original version. It now reads: "In humans, IgG4 responses are believed to be enhanced by the same class switching cytokines as IgA, namely IL-10 and TGF-beta38 in the presence of IL-4." However, some ambiguity remains. I would try to further clarify this statement in order to dissipate any temptation to view IL-10 and TGF-beta as switching factors for both IgA and IgG4. Perhaps a statement like the following could help: "Following induction by IL-4, IgG4 class-switched B cells may enhance IgG4 responses by optimizing their differentiation into IgG4-secreting plasma cells in response to IL-10 and TGF-beta38, two cytokines also required for IgA class switching and IgA production." In general, the term 'class switching' refers to a very specific process involving DNA recombination. Like IL-10, TGF-beta has no TGF-beta-responsive element in the Cgamma4 gene promoter, indicating that these cytokines play no role in IgG4 class switching and only help the IgG4 response through some indirect mechanism. It must be emphasized that class switching from Smu to Sgamma4 has only been demonstrated in B cells exposed to IL-4 or IL-13 and a polyclonal mitogen such as CD40 ligand. This is demonstrated by a vast human literature that includes but is not limited to PMID: 8409416, PMID: 7544371, PMID: 9623503, PMID: 10224410, PMID: 11922943, PMID: 15919378, PMID: 9498752, etc. The PMIDs listed earlier are review articles that were only meant to provide authors with a source of at least some references relevant to human IgG4 class switching. Some of these references are now detailed above and at least one of them (or an equivalent one from the many available) should be included in the revised manuscript.

4) A recent human study, PMID: 32640466, shows the molecular requirements and microbial targets of gut IgG from IBD patients. Also this work should be quoted and briefly discussed.

Reviewer #2 (Remarks to the Author):

The revision of the manuscript has improved it substantially, and the revision and comments from the authors addresses essentially all comments I made on the initial submission. I only have minor issues that need to be addressed in the revised version.

a) I do not know to which extent figure 3D has been revised since the previous version, but I struggle to fully understand it. I think that this is partly due to that the authors sometimes appear to refer to the wrong panels in the results (please go through this carefully to avoid any confusion).

However, I have a couple of suggestion that may improve the figure further.

I do understand the information from the line with black dots - but move the reference dots to the right so that they do not follow in the same line as the other ones to make it more clear that they are references and not data. Also - the figure legends state "... the coloured bar plot ..." but in my version it is black.

The other problems can probably be addressed by a more extensive figure legend.

The middle part of the panel and the results do not seem to correlate fully. The authors state for the Akkermansiaceae family that they see "... a coating frequency of 44% and 40%, respectively." In the diagram it seems that all Akkermansiaceae are coated with either IgA, IgM or both. Please correct or make the description of the plot in the Figure legend more clear as to why there seem to be a discrepancy.

The lower part of the panel shows the bacteria per gram of feces which is easy to undersand. However, I struggle to understand what is presented in the four different rows. The two upper ones are linked and indicated as IgA coated, the two lower as IgM coated. But what does each row actually represent. From the text the IgA+ row should only be from IgA+ controls - why are they dividied into two? If it is double coating (i.e. IgA only vs IgAIgM) - why then two IgM rows - IgM strong was only performed in IgA deficient patients and then there should be no IgA? Or is coated vs non-coated indicated - then please indicate which row is + and which is -. Please clarify in the Figure legend and/or panel so it is clear what each of the four rows represents.

b) The newly added comment about IgG2 is a bit strange. The first sentences state that IgG2 was found in the lumen but did not coat bacteria. The next states that "This finding is in line with a previous study using terminal ileum IgG+ plasmablasts, where on average 36% expressed the IgG2 gene, that were shown to react largely antigen-specific to a panel of commensal and pathogenic gut bacteria .." If the IgG2 plasmablasts in the previous study targeted gut bacteria and IgG2 does not bind gut bacteria here how can it be stated that "This finding is in line with a previous study"?

Oct 9, 2023

Dear reviewers,

First of all, we would once again like to express our gratitude for your time and thorough reviews, and suggestions for improvements of our manuscript. You will find our point-by-point responses below. We hope you will find our revisions satisfactory.

Responses to reviewer 1:

Reviewer #1 (Remarks to the Author):

GENERAL COMMENT

Both clarity and overall insight of this manuscript have improved following revision. Novel aspects are better emphasized and some interpretations clarified. Notwithstanding these improvements and the relative novelty of some findings (i.e., IgG subclass binding profiles and composition of gut microbes from IgA deficient patients; identification of bacteria from these patients dually coated by IgG and IgM), this report transmits a “*dejà vu*” feeling that stems with some overlap with the existing literature (e.g., ref. 13). The lack of antibody binding data from bacteria collected from the intestine further attenuates my enthusiasm for an otherwise interesting and nicely executed study. Some key literature is also missing. That being said, I praise authors for their efforts. Again, the manuscript has certainly improved.

Reply to general comment: We very much appreciate the reviewer’s recognition of the efforts we put into the revision. We are however a bit puzzled by the comment regarding the “*dejà vu*” feeling. We are aware of a previous report concluding that IgM partly rescues IgA binding in IgA deficient subjects by the same authors of ref. 13 (referenced as ref. 8), but here IgG co-binding to fecal bacteria was not reported. The main finding in our study is the observation that IgG often co-bind with IgM to fecal bacteria in IgA deficient subjects. In the referred paper (ref. 13), they take a different approach than us, and the authors explicitly emphasize that they do not identify *in vivo* binding of fecal bacteria by IgG. We do not know why, but it could be due to a difference in methods used to isolate bacteria from fecal material (see further elaboration on this part below). Instead, the authors of ref 13 perform *ex vivo* incubation between bacteria and serum IgG, showing that serum IgGs can bind fecal bacteria. There are no data presented in relation to co-incubation of bacteria with both serum IgG and IgM, and therefore our identification of a high frequency of *in vivo* co-binding of IgM and IgG to fecal bacteria in IgA deficiency has not been shown in ref 13, or in the literature so far. Additionally, the authors argue that IgGs cannot be transported across the gut barrier. This is in contrast to reports demonstrating active transport of IgGs via the neonatal Fc receptor, and the findings by others and us of high fecal IgG levels.

In fact, we have been puzzled by the discrepancy between the results of the two studies, and therefore went on to compare the methods used in ref 13 and our study. Our study was initiated in 2016, and method choices were therefore not directed by those in ref. 13. When comparing the

methods for preparation of fecal pellets in the two studies, we realized that they were quite different, and that the procedure used in ref. 13 was originally optimized to be used for bacterial proteomics, including extensive centrifugation steps. Whether this may lead to the loss of the IgG-coated bacteria can only be speculated at this point, but since others, as referenced in the manuscript (ref. 16 and 17 (humans)), have also identified IgG-coating of gut bacteria in humans to the same levels as identified in our study, we would take the liberty to argue that the present study findings cannot, at least in our opinion, be a déjà vu of ref 13, as the two reports, to a large extent, have opposing conclusions.

It is therefore difficult from our perspective to understand how our findings can transmit a ‘déjà vu’ feeling to existing literature, besides the common focus in existing literature on IgA deficiency and IgM or IgA coating of fecal bacteria.

There is also a comment related to the lack of data on antibody binding from bacteria collected in the intestine. Before considering to move on to endoscopic procedures for collection of intestinal bacterial samples, we first wanted to get an overview of where in the intestine, plasma cells with IgM and IgG expression are located. We did this by analysing a publically available single cell dataset with sampling at six sites along the human intestinal tract (PMID: 37468586), focusing our analysis on the expression of Ig heavy chains in plasma cells. As seen in the inserted figure below, the dominant intestinal location of Ig heavy chain expressing plasma cells varies depending on the

Ig subtype. While expression of *IGHG1* is highest in plasma cells in the ileum and the distal part of the colon, the expression of *IGHG2* is highest in the transverse colon, *IGHG3* in the jejunum, and *IGHG4* in the ileum. *IGHA2* is primarily expressed in colonic plasma cells, as expected based on literature (PMID: 17227687), whereas *IGHM* is expressed in small intestinal plasma cells. Co-

binding with IgG1 or IgG4 and IgM may therefore be possible to pick up within the ileum mucosa, whereas co-binding of bacteria with IgG2 or IgG3 and IgM would be more spurious as IgG2 and IgG3 are less frequently identified to bind to fecal bacteria at all (Fig. 6d,f), and a considerable fraction of IgG2 is found unbound in feces (Fig. 6e). According to the intestinal distribution of Ig expression in plasma cells, one would therefore have the highest chance of identifying co-binding of IgG and IgM to bacteria collected within the mucosa of the ileum. As ileoscopy of healthy individuals carries a risk with it, we have decided not to address this comment as part of this manuscript revision, as we envisage that the added insights from examining antibody co-binding profiles in ileal mucosa derived bacteria would not justify the discomfort that an ileoscopy would inflict on the IgA deficient subjects (and their partners). Bacteria collected from luminal intestinal content in the distal colon would not add more information than the one we already obtained from feces.

We have updated the manuscript with some of the proposed references, as pointed out below.

SPECIFIC COMMENTS

1) Some of the findings of this work are incremental at best. This is the case of data showing minimal IgG binding to fecal bacteria in healthy controls. In their rebuttal letter, authors state: "There is presently only a limited number of reports demonstrating direct IgG-coating of gut bacteria (PMID: 31771976; PMID: 30606251; PMID: 15201580; PMID: 15201580) ..." Aside from inadvertently duplicating PMID: 15201580, authors omit PMID: 30876876, which compares IgG+ fecal bacteria from household healthy controls with IgG+ fecal bacteria from ulcerative colitis patients. PMID: 30876876 should be quoted and findings from its Fig. 1B briefly discussed, also in relationship to PMID: 30554723.

Reply 1: We thank the reviewer for identifying our duplication mistake of PMID: 15201580 in the first rebuttal letter. Also, thanks a lot for pointing out PMID: 30876876, which we did not identify as relevant in our literature research, as it mainly focuses on mouse data. However, we acknowledge that their Fig. 1b is showing data for total IgG+ coating of fecal bacteria in ulcerative colitis (UC) patients vs. healthy controls (n=6 for each), and reports enhanced IgG+ coating in UC patients, while total IgA+ coating is not increased. Their data on total IgG+ coating in healthy controls are in the range of those we find in our study. We have now included PMID: 30876876, and revised the sentence: Page 5, line 16: healthy human adults¹⁶ and infants¹⁷, and reported in mice¹².

It is unfortunately not clear to us why the reviewer would like to discuss these findings in relation to the data in PMID: 30554723 (ref. 13), as ref. 13 does not find similar amounts of IgG+ coated bacteria, as we have already mentioned earlier in the same sentence inserted above (Page 5, line 16). However, we speculate that the discrepancy between ref. 13 and the other studies may relate to the different method used for fecal pellet preparation in ref. 13 as compared to the other studies, as it was originally optimized to be used for bacterial proteomics, including extensive centrifugation steps. Whether this may lead to the loss of the IgG-coated bacteria can only be speculated at this point, but since others, as referenced in the manuscript (ref. 16 (humans)), have identified the same level of overall IgG-coating of gut bacteria in humans as seen in our study, we would take the liberty to argue that the present study findings cannot, at least in our opinion, be seen as incremental and a *déjà vu* of ref 13, as the two reports, to a large extent, have opposing conclusions.

In order to bring the discrepancy of the studies to the attention of the reader, we have now included the following sentence in the discussion (Page 13, line 12): ‘A few previous studies in humans have identified IgG bacterial coating mounting to the same level as in the current study (median: 1.7%) (adults¹⁶, infants¹⁷), while another study identified minute levels of bacterial IgG coating in humans (median: 0.03%)¹³. A difference in the method used for fecal pellet preparation, where Fadlallah *et al.*¹³ employed a method optimized for bacterial proteomics is speculated to explain the discrepancy’.

2) In PMID: 30554723 (ref. 13), there is a virtual lack of IgG+ bacteria in the stool from healthy individuals, which contrasts with the much higher frequency of these bacteria in the present study. Can authors elaborate on possible reasons behind this striking discrepancy?

Reply 2: This point has been addressed in our reply to the general comment and in reply 1. Briefly, we envisage the discrepancy may rely on the very different approach used for bacterial pellet isolation in ref. 13, where some of the coated bacteria may be lost during the applied gradient centrifugation steps, as compared to our study and the two other studies of total IgG+ fecal bacterial coating (ref. 16 and 17) that all use protocols for fecal bacterial isolation similar to the one we used for ELISA on fecal bacteria.

3) Line 22 of the revised Discussion is more accurate than its original version. It now reads: “In humans, IgG4 responses are believed to be enhanced by the same class switching cytokines as IgA, namely IL-10 and TGF-beta38 in the presence of IL-4.” However, some ambiguity remains. I would try to further clarify this statement in order to dissipate any temptation to view IL-10 and TGF-beta as switching factors for both IgA and IgG4. Perhaps a statement like the following could help: “Following induction by IL-4, IgG4 class-switched B cells may enhance IgG4 responses by optimizing their differentiation into IgG4-secreting plasma cells in response to IL-10 and TGF-beta38, two cytokines also required for IgA class switching and IgA production.” In general, the term ‘class switching’ refers to a very specific process involving DNA recombination. Like IL-10, TGF-beta has no TGF-beta-responsive element in the Cgamma4 gene promoter, indicating that these cytokines play no role in IgG4 class switching and only help the IgG4 response through some indirect mechanism. It must be emphasized that class switching from Smu to Sgamma4 has only been demonstrated in B cells exposed to IL-4 or IL-13 and a polyclonal mitogen such as CD40 ligand. This is demonstrated by a vast human literature that includes but is not limited to PMID: 8409416, PMID: 7544371, PMID: 9623503, PMID: 10224410, PMID: 11922943, PMID: 15919378, PMID: 9498752, etc. The PMIDs listed earlier are review articles that were only meant to provide authors with a source of at least some references relevant to human IgG4 class switching. Some of these references are now detailed above and at least one of them (or an equivalent one from the many available) should be included in the revised manuscript.

Reply 3: Thanks a lot for, once again, pointing at this important part. We have now changed the part on page 15 line 10: ‘Following induction by IL-4³⁸ or IL-13³⁹, IgG4 class-switched B cells may enhance IgG4 responses by optimizing their differentiation into IgG4-secreting plasma cells in

response to IL-10 and TGF-beta⁴⁰⁻⁴³, two cytokines also required for IgA class switching and IgA production.'

4) A recent human study, PMID: 32640466, shows the molecular requirements and microbial targets of gut IgG from IBD patients. Also this work should be quoted and briefly discussed.

Reply 4: It is delicate to decide on the correct references for a study, and in this instance we do not agree that the suggested study is relevant to cite. We will try to explain in the following: We agree that findings in PMID: 32640466 are of importance in the search for poly- vs. single bacterial response patterns amongst IgA and IgG clones derived from human intestinal plasma cells. However, since the human-relevant part of their data are derived from *ex vivo* incubation of the monoclonal Abs added to non-Ig-ablated human fecal bacteria, we deliberately chose not to reference this paper, as it, in our opinion, does not compare to our data that identify *in vivo* coating profiles. In addition, we do not consider their data on binding of human IgA and IgG monoclonals from healthy controls and Crohn's disease (CD) patients to fecal microbiota from SPF Rag2-deficient mice to be relevant to cite amongst the few articles that the word limitations allow for. This decision is based on compositional differences between mouse and human fecal bacteria. In this context one could speculate that their findings of an enhanced binding of human IgG monoclonals from CD patients compared to healthy controls to murine fecal bacteria could be due to the fact that aerobic strains are more alike in mouse and man (than their anaerobic counterparts), and that IgG monoclonals from CD patients may be better at targeting aerobic strains (CD patients have a gut microbiota depleted from many anaerobic strains). We therefore do not agree that it would strengthen the present paper (that primarily focuses on IgM and IgG co-binding to fecal bacteria in IgAD) to initiate such a discussion, which was why we did not cite the findings of PMID: 32640466 in our original submission.

Reviewer #2 (Remarks to the Author):

The revision of the manuscript has improved it substantially, and the revision and comments from the authors addresses essentially all comments I made on the initial submission. I only have minor issues that need to be addressed in the revised version.

Reply to intro: Thanks a lot. We very much appreciate the reviewer's recognition of the efforts we put into the revision.

a) I do not know to which extent figure 3D has been revised since the previous version, but I struggle to fully understand it. I think that this is partly due to that the authors sometimes appear to refer to the wrong panels in the results (please go through this carefully to avoid any confusion).

However, I have a couple of suggestion that may improve the figure further.

I do understand the information from the line with black dots - but move the reference dots to the right so that they do not follow in the same line as the other ones to make it more clear that they

are references and not data. Also - the figure legends state "... the coloured bar plot ..." but in my version it is black.

The other problems can probably be addressed by a more extensive figure legend.

The middle part of the panel and the results do not seem to correlate fully. The authors state for the Akkermansiaceae family that they see "... a coating frequency of 44% and 40%, respectively." In the diagram it seems that all Akkermansiaceae are coated with either IgA, IgM or both. Please correct or make the description of the plot in the Figure legend more clear as to why there seem to be a discrepancy.

The lower part of the panel shows the bacteria per gram of feces which is easy to understand. However, I struggle to understand what is presented in the four different rows. The two upper ones are linked and indicated as IgA coated, the two lower as IgM coated. But what does each row actually represent. From the text the IgA+ row should only be from IgA+ controls - why are they divided into two? If it is double coating (i.e. IgA only vs IgAIgM) - why then two IgM rows - IgM strong was only performed in IgA deficient patients and then there should be no IgA? Or is coated vs non-coated indicated - then please indicate which row is + and which is -. Please clarify in the Figure legend and/or panel so it is clear what each of the four rows represents.

Reply to a):

First of all, we are sorry that the revision of Fig. 3d was not sufficiently described in the general comments provided on page 1 of the point-by-point response to the first revision. As stated on page 1 for Fig. 3 revisions, we, during the reanalysis for the first revision, recognized that the cutoff filters were too conservative, and changed the cutoff in filters for ASV abundance to 0.001% from earlier 0.01%. This change resulted in identification of 5 additionally coated bacterial families (now 23, previously 18). Besides some minor shuffling of legends, Fig. 3d was only updated to now include the 23 bacterial families.

According to your other comments, and to make Fig. 3d easier to read and understand, we have now made a few minor changes to Fig. 3d. Moved the 'dot' legend, as proposed, reduced the distance between the squares in the lower panel of the figure + reintroduced the legend at the right hand side of the lower panel, and moved the description of these to the right place in the figure text. We have also introduced headers for each of the three subpanels for further clarification. The reference in the figure legend to "... the coloured bar plot" has now been changed to the bar plot.

The manuscript text has been thoroughly updated on page 8 of the manuscript to revise the mistaken Fig. 3 subfigure references.

Re. "The middle part of the panel and the results do not seem to correlate fully. The authors state for the Akkermansiaceae family that they see "... a coating frequency of 44% and 40%, respectively." In the diagram it seems that all Akkermansiaceae are coated with either IgA, IgM or both. Please correct or make the description of the plot in the Figure legend more clear as to why there seem to be a discrepancy".

Thanks a lot for pointing out this important point that needed further clarification. We have now revised it, Page 8 line 6 and it now reads: 'Members of the *Akkermansiaceae* family accounted for

only 1.6% of all coated bacterial taxa ($n = 26$, Fig 3b), but a relatively high number of these taxa was coated with IgA or IgM (3.47×10^8 bacteria/g feces with IgA in IgA+ and 1.75×10^8 bacteria/g feces with IgM in IgA- subjects, Fig 3d (lower panel)), representing an average coating frequency per subject of 44% and 40%, respectively. This should be seen in contrast to the fact that all *Akkermansiaceae* ASVs were identified to be coated across all of the subjects (Fig 3d, middle panel).’ We have also now clarified in other text parts related to fig. 3 that we refer to averages across all subjects, and removed the mistaken references to fig. 3d, middle panel.

b) The newly added comment about IgG2 is a bit strange. The first sentences state that IgG2 was found in the lumen but did not coat bacteria. The next states that "This finding is in line with a previous study using terminal ileum IgG+ plasmablasts, where on average 36% expressed the IgG2 gene, that were shown to react largely antigen-specific to a panel of commensal and pathogenic gut bacteria .." If the IgG2 plasmablasts in the previous study targeted gut bacteria and IgG2 does not bind gut bacteria here how can it be stated that "This finding is in line with a previous study"?

Reply to b:

Sorry, we see your point. It needs a bit of more clarification, and a rewrite. We have now rephrased it as follows, page 12 line 2: ‘Although IgG2 has long been recognized to be induced in response to microbial pathogens possessing repeating carbohydrate antigens (encapsulated microbes), its possible role in binding to non-bacterial members, such as viruses, within the human intestinal tract has not been addressed thoroughly. An earlier study identified that terminal ileum IgG+ plasmablasts (IgG1: 51.8%, IgG2: 35.9%, IgG3: 2.5% and IgG4: 9.9%) react largely antigen-specific to a panel of commensal and pathogenic gut bacteria, and virus-specific binding was also detected²⁴. A specific role for IgG2 in virus-binding was however not specified.’

On behalf of all authors,
Susanne Brix

REVIEWERS' COMMENTS

Reviewer #1 (Remarks to the Author):

The authors addressed my last comments. The manuscript has significantly improved and should be published.

Reviewer #2 (Remarks to the Author):

I am really sorry - but I do still struggle with the lower part of figure 3D, and I guess that this will be true for many other readers as well.

First, in the figure legend it is stated for panel 3d.

"The lower heatmap displays the mean actual number of Ig-coated bacterial families per g feces in IgA+ and IgA- subjects, respectively."

To me, this indicates that I should see data from IgA+ individuals and IgA- individuals separately. For the uppermost row there is an indication as IgA+ to the right (which I presume is bacteria that only bind to IgA. However, the legend says

"IgA+" marks the bacterial families that are only coated in IgA+ subjects".

So this is the amount of bacteria binding only IgA+ in IgA+ subjects. Or does it mean bacteria with any coating at all (IgA/IgM). Or is it the bacterial families that IgA coated in IgA+ individuals that were not IgM coated in IgA- individuals? Looking in the results section it says

"all IgA coated bacteria in IgA+ subjects and all IgM coated bacteria in IgA- subjects"

were sorted, so I guess that last interpretation is right (although I am not absolutely sure)?

I then do the same for the lowest row - and read in the legend

"IgM+" marks the bacterial families only coated in IgA- subjects"

and assume that these were sorted based on IgM coating in IgA- individuals, and analogously would then be bacterial species that were coated with IgM in IgA- individuals but not by IgA in IgA+ individuals (although possibly coated with IgM in them?).

That leaves the two IgA+/IgM+ rows in the middle is described as

"IgA+/IgM+" indicates that these bacterial families are coated in both IgA+ and IgA- subjects".

If my interpretation is right, the upper of these two would then come from IgA+ individuals, be coated with IgA, and belong to families also being IgM coated in IgA- individuals, and the lower IgM coated in IgA- individuals, and belonging to families that were IgA coated in IgA+ individuals. Right?

If this is the correct interpretation, I would suggest that the authors to make this clearer in the figure. One suggestion would be to move the upper two rows so that they touch each other and the lower so they touch each other but leave a small gap in the middle - thus bacterial species from one type of donor would then be together. I would then not use "IgA coated" and "IgM coated" but rather "IgA coated in IgA+ individuals" and "IgM coated in IgA- individuals". Finally, instead of IgA+, IgA+IgM+ and IgM+ I would use the term "unique" for IgA+ and IgM+ and "shared" for IgA+IgM+. Of course, the should then also need to figure legend to be modified. I think that figure 4a is much more clearer and easier to understand, and in this case the donors are kept divided (although possibly donor info should be added there as well, i.e. IgA+ donor and IgA-donor).

I am happy with other solutions, but in its current form I think that most readers would otherwise interpret "IgA+" as bacteria only coated with IgA, "IgM+" as bacteria only coated with IgM and "IgA+/IgM+" as bacteria coated with both IgM and IgA. And would also not understand from which donors each set of data came from. It actually took me over an hour to get to the right conclusion after very carefully reading the results, the figure legend and looking at the figure (and I am quite honestly still not absolutely sure that this is the right conclusion).

Nov 7, 2023

Dear reviewer,

Thanks again for your very thorough peer-review, and for spotting the below problem.

Reviewer #2 (Remarks to the Author):

I am really sorry - but I do still struggle with the lower part of figure 3D, and I guess that this will be true for many other readers as well.

First, in the figure legend it is stated for panel 3d.

"The lower heatmap displays the mean actual number of Ig-coated bacterial families per g feces in IgA+ and IgA- subjects, respectively."

To me, this indicates that I should see data from IgA+ individuals and IgA- individuals separately. For the uppermost row there is an indication as IgA+ to the right (which I presume is bacteria that only bind to IgA. However, the legend says

"“IgA+” marks the bacterial families that are only coated in IgA+ subjects".

So this is the amount of bacteria binding only IgA+ in IgA+ subjects. Or does it mean bacteria with any coating at all (IgA/IgM). Or is it the bacterial families that IgA coated in IgA+ individuals that were not IgM coated in IgA- individuals? Looking in the results section it says

"all IgA coated bacteria in IgA+ subjects and all IgM coated bacteria in IgA- subjects"

were sorted, so I guess that last interpretation is right (although I am not absolutely sure)? Yes the last sentence is correct.

I then do the same for the lowest row - and read in the legend

"“IgM+” marks the bacterial families only coated in IgA- subjects"

and assume that these were sorted based on IgM coating in IgA- individuals, and analogously would then be bacterial species that were coated with IgM in IgA- individuals but not by IgA in IgA+ individuals (although possibly coated with IgM in them?). Correct.

That leaves the two IgA+/IgM+ rows in the middle is described as

"“IgA+/IgM+” indicates that these bacterial families are coated in both IgA+ and IgA- subjects".

If my interpretation is right, the upper of these two would then come from IgA+ individuals, be

coated with IgA, and belong to families also being IgM coated in IgA- individuals, and the lower IgM coated in IgA- individuals, and belonging to families that were IgA coated in IgA+ individuals. Right? Yes, correct.

If this is the correct interpretation, I would suggest that the authors to make this clearer in the figure. One suggestion would be to move the upper two rows so that they touch each other and the lower so they touch each other but leave a small gap in the middle - thus bacterial species from one type of donor would then be together. I would then not use "IgA coated" and "IgM coated" but rather "IgA coated in IgA+ individuals" and "IgM coated in IgA- individuals". Finally, instead of IgA+, IgA+IgM+ and IgM+ I would use the term "unique" for IgA+ and IgM+ and "shared" for IgA+IgM+. Of course, the should then also need to figure legend to be modified. I think that figure 4a is much more clearer and easier to understand, and in this case the donors are kept divided (although possibly donor info should be added there as well, i.e. IgA+ donor and IgA- donor).

I am happy with other solutions, but in its current form I think that most readers would otherwise interpret "IgA+" as bacteria only coated with IgA, "IgM+" as bacteria only coated with IgM and "IgA+/IgM+" as bacteria coated with both IgM and IgA. And would also not understand from which donors each set of data came from. It actually took me over an hour to get to the right conclusion after very carefully reading the results, the figure legend and looking at the figure (and I am quite honestly still not absolutely sure that this is the right conclusion).

Reply: We have now revised Fig. 3d as suggested, and modified the figure text accordingly to: I9 p29: '(d) The heatmap in the lower panel displays the average actual number of Ig-coated bacterial families per g feces in IgA+ and IgA- subjects, respectively. Grey boxes represent bacterial families with no identified Ig-coating. The bar plot in the middle panel summarizes the relative distribution of coating types for each of the bacterial families. The dot plot in the upper panel displays the average relative abundance of each bacterial family in bulk feces. (a, b, c, d), 'IgA coated' indicates that bacteria are found in the sorted IgA-coated fraction from IgA+ subjects (unique for IgA+), but not in the sorted IgM-coated fraction from IgA- subjects, and vice versa for 'IgM coated' (unique for IgA-); 'IgA+/IgM+' indicates that bacteria are found to be coated with IgA in IgA+ subjects and with IgM in IgA- subjects (shared). 'Strictly non-coated' indicates that bacteria were only identified in non-coated fractions'. The text part within the related results section (I26, p7 – I18, p8) has been updated to more clearly specify when we refer to the upper, middle or lower panel in Fig. 3d. We are very sorry for this confusion, and the time you have spent on this.

On behalf of the authors,
Carsten Eriksen and Susanne Brix